# Hakai is required for stabilization of core components of the m6A mRNA methylation machinery

Praveen Bawankar[1,11], Tina Lence [2,10,11], Chiara Paolantoni[3,11], Irmgard U. Haussmann[4,5], Migle Kazlauskiene [6], Dominik Jacob[1], Jan B. Heidelberger[2], Florian M. Richter[1], Mohanakarthik P. Nallasivan[4], Violeta Morin[2], Nastasja Kreim[7], Petra Beli [2,8], Mark Helm [1], Martin Jinek [6], Matthias Soller [4,9✉] & Jean-Yves Roignant [1,3✉]

$N^6$-methyladenosine (m6A) is the most abundant internal modification on mRNA which influences most steps of mRNA metabolism and is involved in several biological functions. The E3 ubiquitin ligase Hakai was previously found in complex with components of the m6A methylation machinery in plants and mammalian cells but its precise function remained to be investigated. Here we show that Hakai is a conserved component of the methyltransferase complex in *Drosophila* and human cells. In *Drosophila*, its depletion results in reduced m6A levels and altered m6A-dependent functions including sex determination. We show that its ubiquitination domain is required for dimerization and interaction with other members of the m6A machinery, while its catalytic activity is dispensable. Finally, we demonstrate that the loss of Hakai destabilizes several subunits of the methyltransferase complex, resulting in impaired m6A deposition. Our work adds functional and molecular insights into the mechanism of the m6A mRNA writer complex.

[1] Institute of Pharmaceutical and Biomedical Sciences, Johannes Gutenberg-University Mainz, Mainz, Germany. [2] Institute of Molecular Biology (IMB), Mainz, Germany. [3] Center for Integrative Genomics, Génopode Building, Faculty of Biology and Medicine, University of Lausanne, Lausanne, Switzerland. [4] School of Biosciences, College of Life and Environmental Sciences, University of Birmingham, Birmingham, UK. [5] Department of Life Science, Faculty of Health, Education and Life Sciences, Birmingham City University, Birmingham, UK. [6] Department of Biochemistry, University of Zurich, Zurich, Switzerland. [7] Bioinformatics core facility, Institute of Molecular Biology (IMB), Mainz, Germany. [8] Institute of Developmental Biology and Neurobiology (IDN), Johannes Gutenberg-Universität, Mainz, Germany. [9] Birmingham Centre for Genome Biology, University of Birmingham, Birmingham, UK. [10]Present address: Institute for Molecular Infection Biology (IMIB), Faculty of Medicine, University of Würzburg, Würzburg, Germany. [11]These authors contributed equally: Praveen Bawankar, Tina Lence, Chiara Paolantoni. ✉email: M.Soller@bham.ac.uk; jean-yves.roignant@unil.ch

N[6]-methyladenosine (m6A) is one of the most abundant and well-studied mRNA modifications in eukaryotes[1–4]. This modification plays a central role in almost every aspect of mRNA metabolism, and is essential for several biological processes such as cell differentiation[1–8], DNA repair[9], circadian rhythm[10–12], neurogenesis[13] and sex determination[14–16], among others. Its dysregulation in human is associated with numerous diseases, including metabolic alteration[17], neuronal disorders[18,19] and various types of cancers[20–22]. The downstream effects of m6A are generally mediated by YTH domain RNA-binding proteins, known as "m6A readers" that preferentially bind m6A modified RNAs and affect their fate[3].

m6A on mRNA is deposited co-transcriptionally by a conserved multiprotein complex that can be divided into two stable sub-complexes: the heterodimer METTL3/METTL14 also known as m6A-METTL Complex (MAC) that contains the catalytic activity, and the m6A-METTL Associated Complex (MACOM) that is required for full MAC activity and includes WTAP (Fl(2) d), VIRMA (Virilizer), RBM15/RMB15B (Spenito) and ZC3H13 (Flacc)[23–25]. More recently, another factor named HAKAI was found associated with components of MACOM in plants and human cells and required to maintain m6A level[26,27].

The precise function of MAC and MACOM components has been the subject of intense research over the past years. Structural studies revealed that METTL3 and METTL14 form a stable heterodimer[28–30] and that METTL3 is the only factor that contains the catalytic activity since METTL14 is unable to bind the methyl group donor S-Adenosylmethionine. Nevertheless, METTL14 is essential to support the interaction of the complex with its RNA targets and to enhance METTL3 activity. WTAP was shown to stabilize the interaction between METTL3 and METTL14 and to recruit the METTL3/METTL14 heterodimer into the nuclear speckles[15,31]. RBM15/RBM15B is an RNA-binding protein that recognizes U-rich sequences on the mRNA and is suggested to recruit the m6A machinery in close proximity to these sites[32]. Furthermore, VIRMA was proposed to facilitate selective m6A installation near the stop codon and in the 3′ UTR of mRNAs through its interaction with polyadenylation cleavage factors CPSF5 and CPSF6[27]. Lastly, four recent studies identified ZC3H13 as part of MACOM[24,27,33,34]. ZC3H13 was found to stabilize the interaction between WTAP and RBM15 in mouse embryonic stem cells as well as in flies and to contribute to the localization of the writer complex to the nucleus. The only m6A writer component whose function is still poorly explored is HAKAI.

HAKAI, also known as CBLL1, is a RING-finger type E3 ubiquitin ligase that mediates ubiquitination and subsequent endocytosis of the E-cadherin complex, leading to cell-cell adhesion loss and increased cell motility[35]. HAKAI can also regulate cell proliferation in an E-cadherin-independent manner by affecting the ability of the PTB-associated splicing factor to bind some of its RNA targets[36]. During the last decades, HAKAI has been mostly studied in the context of epithelial-mesenchymal transitions and cancer progression[37]. However, as aforementioned, it became increasingly clear that HAKAI is also a component of the m6A biogenesis machinery in vertebrates as well as in plants. HAKAI was identified as one of the strongest WTAP interactors in mammalian cells[38] and as part of an evolutionary conserved protein complex including WTAP, VIRMA and ZC3H13[39]. Furthermore, Hakai mutant in Arabidopsis thaliana displayed mild developmental defects together with reduced m6A levels[26]. Recently, we identified Hakai among the top enriched proteins in our Spenito (Nito) interactome in Drosophila S2R+ cells, suggesting its evolutionary conserved role within the m6A pathway[24].

Here we report that Hakai is a conserved member of MACOM and is essential for m6A deposition in flies. In line with its role in the m6A pathway, Hakai functions in the sex determination pathway and mediates splicing of Sex lethal. Moreover, its depletion results in altered gene expression and splicing changes that resemble the loss of other MACOM components. We find that Hakai in flies encodes short and long protein isoforms that display distinct subcellular localization. Its ubiquitin ligase domain is required for homodimerization and interaction with other MACOM components. Finally, we show that Hakai removal leads to a severe reduction of Virilizer (Vir), Fl(2)d and Flacc protein levels, indicating that Hakai is essential for maintaining the stability of MACOM components.

## Results

**Drosophila Hakai is a conserved MACOM subunit.** We previously found Hakai among highly enriched proteins in Nito and RBM15 interactomes in Drosophila S2R+ and mouse ES cells, respectively, suggesting it is a conserved member of MACOM (Fig. 1a)[24]. In Drosophila, Hakai can generate four protein isoforms via alternative splicing: the two short and two long isoforms differ in the extension of the C-terminal region as a result of intron retention and in the length of the second exon due to alternative 3′ splice sites in the first intron (Fig. 1b). All proteins share a RING-type E3 ubiquitin ligase domain and an adjacent C2H2-like zinc finger. This region is highly conserved and was shown in mammals to be required for Hakai dimerization and formation of the so-called "Hakai phosphotyrosine-binding domain" (HYB domain) (Supplementary Fig. 1)[40]. Expression of RNA isoforms was monitored during development by real-time quantitative PCR (Supplementary Fig. 2). Both short and long isoforms greatly overlapped with the m6A profile, showing high enrichment during early embryogenesis and in ovaries, which is consistent with the transcript distribution of the other subunits of the m6A methyltransferase complex[15,24]. In addition, the long isoforms were particularly elevated in males, suggestive of a possible function during spermatogenesis. Intriguingly, when we examined their subcellular localization in S2R+ cells we found that the short isoform (302 aa) was present exclusively in the cytoplasm with a strong signal at the cellular periphery, in contrast to the long isoform (473 aa) that was predominantly nuclear (Fig. 1c). A similar result was obtained in BG3 cells, which are cells derived from larval brains (Supplementary Fig. 3a). Furthermore the long isoform colocalized to sites of transcription on polytene chromosomes of salivary glands (Supplementary Fig. 3b). Altogether, since m6A is deposited co-transcriptionally, these results suggest that only the long isoform may be relevant with regards to m6A biogenesis.

To address if Hakai indeed interacts with MACOM components in Drosophila, we cloned Hakai cDNA encoding long (473 residues) isoform in a Myc-GFP-tagged expression vector and transfected the construct in Drosophila S2R+ cells to carry out Myc pull-down assay followed by mass spectrometry analysis. We found that the tagged Hakai was able to immunoprecipitate all MACOM components, even though the interaction with Nito was just below the cutoff (Fig. 1d, Supplementary Data 1). Furthermore, Mettl3 and Mettl14 were absent from the precipitated proteins, confirming our previous findings that the interaction between MAC and MACOM is either weak or transient.

To confirm these results, we performed co-immunoprecipitation assays in S2R+ cells. We found that Hakai can interact with other MACOM components, irrespective of the presence of RNA (Fig. 1e, f). Further confirmation of the interaction between Hakai and Fl(2)d/Nito was obtained via

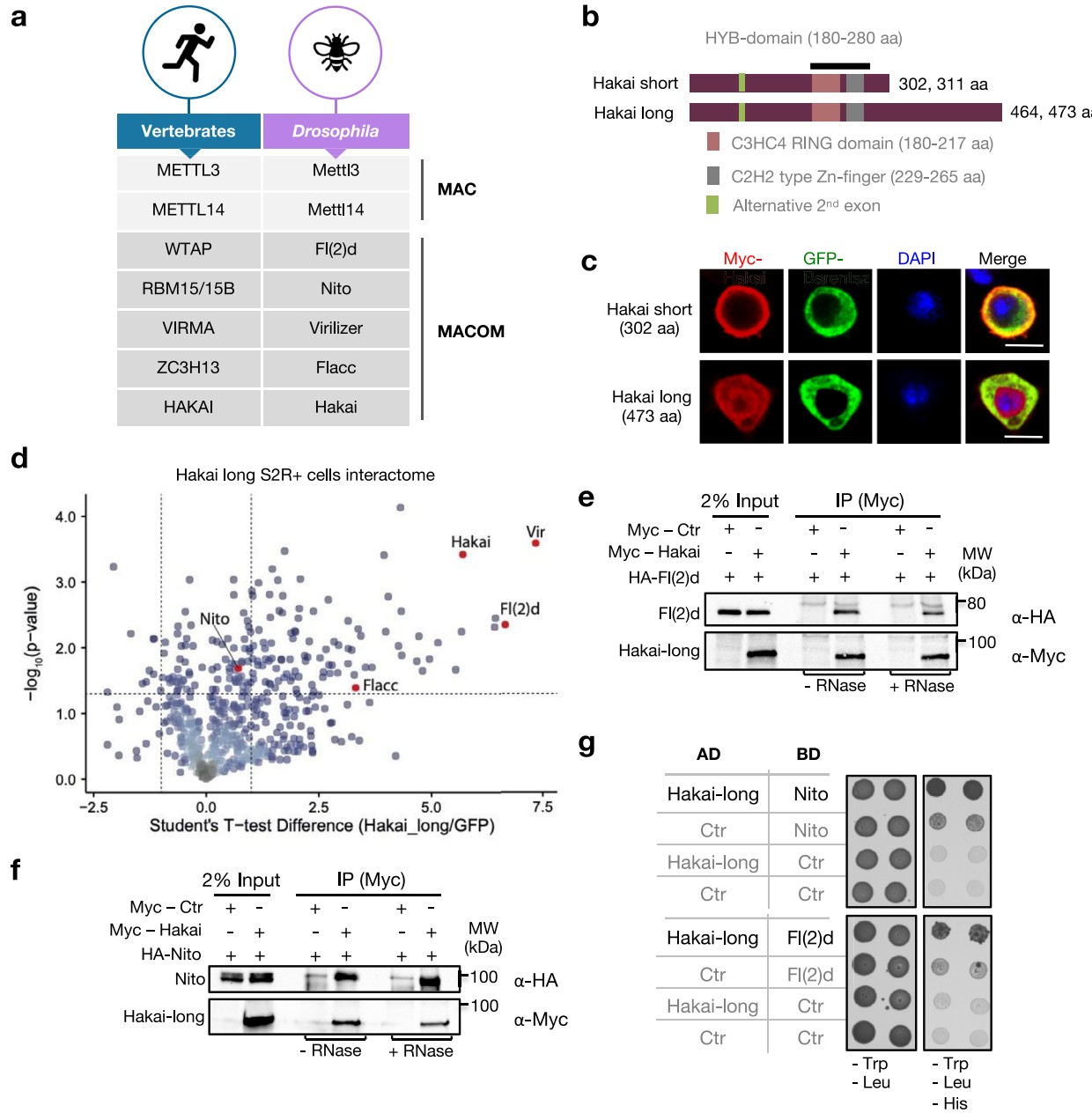

**Fig. 1 Hakai interacts with MACOM components. a** Table indicating the names of MAC and MACOM components in Vertebrates and *Drosophila*. **b** Schematic of the Hakai short and long protein isoforms depicting the RING domain (pink), the Zn-finger (grey) and the HYB domain (black). The green domain indicates the alternative region of the second exon that is included in isoforms 311 aa and 473 aa. **c** Immunostaining of Myc-tagged Hakai short and long protein isoforms overexpressed in S2R+ cells. GFP-tagged Barentsz protein served as a cytoplasmic marker. DAPI staining is shown in blue. The short Hakai isoform localizes strictly to the cytoplasm, whereas the long isoform localizes to both cellular compartments with the enrichment in the nucleus. Scale bars, 10 μm. **d** Identification of proteins interacting with GFP-Hakai-long in S2R+ cells based on label-free analysis of two replicate experiments analyzed by quantitative MS-based proteomics. HAKAI immunoprecipitates alongside of all MACOM components. MACOM component proteins are highlighted in red. A complete list of quantified proteins can be found in Supplementary Data 2. **e**, **f** Co-immunoprecipitation experiments were carried out with lysates prepared from S2R+ cells transfected with Myc-tagged Hakai (long isoform) and HA-tagged Nito or Fl(2)d. In control lanes, S2R+ cells were transfected with Myc alone and an identical HA-containing protein. Extracts were immunoprecipitated with anti-Myc antibody and immunoblotted using anti-Myc and anti-HA antibodies. Two percent of input was loaded. The same experiment was repeated in the presence of RNase T1. Nito and Fl(2)d interact with Hakai in an RNase-independent manner. Blots shown are representative of one biological replicate. **g** Yeast-two-hybrid assay to investigate Hakai interaction with Nito and Fl(2)d. Proteins were cloned in yeast expression vectors and fused with either Gal4-DNA binding domain (BD) or Gal4-DNA activation domain (AD). Indicated combinations of vectors were co-expressed in yeast and empty vectors encoding only activation or binding domain were used as control (Ctr). Recovered colonies were spotted on plates lacking Leucine and Tryptophan (-Leu, -Trp) as well as selection plates lacking amino acids Leucine, Tryptophan and Histidine (-Leu, -Trp, -His). AD-Hakai long isoform interacts with BD-Fl(2)d and BD-Nito. Source data for Western blots and the yeast-two-hybrid assay are provided as a Source Data file.

yeast-two-hybrid (Y2H) assay (Fig. 1g). However, interaction between Hakai and Flacc was not detected in this assay and Vir could not be successfully expressed in yeast, probably due to its large size. Furthermore, in agreement with the human HAKAI structure[40], we observed that the Hakai fly orthologue could homodimerise, as well as Fl(2)d and Nito, which was not reported previously (Supplementary Fig. 4a, b).

Altogether our data indicate that Hakai is a component of the m[6]A machinery in flies, that it interacts with MACOM components, and that several subunits within this complex likely exist in more than one copy.

**Vir functions as a scaffold for MACOM component interaction.** To gain better mechanistic insights of Hakai within MACOM, we precisely mapped the intermolecular interactions between the different components of the complex. To do so, we performed co-immunoprecipitation experiments in the presence of RNase A using full length as well as fragments of MACOM subunits in different combinations (Fig. 2a). Results from these experiments revealed that Hakai N-terminal region (residues 1-295, common to both isoforms) interacts with the N-terminal region of Vir (residues 1–130) (Fig. 2b, c), while the middle region and C-terminus of Fl(2)d (residues 125-536) binds Vir C-terminal (residues 1501–1854) (Fig. 2d, e). The same region of Fl(2)d was also required for homodimerisation (Supplementary Fig. 4a). These data suggest that Vir may mediate the interaction between Fl(2)d and Hakai. To address this possibility, we tested whether Hakai and Fl(2)d still interact in the absence of Vir. The co-immunoprecipitation assay revealed that the interaction was strongly compromised upon *vir* KD (Fig. 2f and Supplementary Fig. 5a). Similarly, the interaction between Hakai and Nito was dramatically reduced (Supplementary Fig. 5a, b). In contrast, the lack of Hakai did not interfere with the association of Fl(2)d with Nito or Vir (Supplementary Fig. 5c–e). Therefore these results suggest that Vir likely stabilize the interaction between Fl(2)d, Hakai and Nito (Fig. 2g).

We next explored whether these interactions are conserved in the human MACOM. To this end, we overexpressed several human MACOM subunits (Supplementary Fig. 6a) in HEK293T-cells and performed co-imunoprecipitation experiments in the presence of RNase A. We found that the interaction between HAKAI and the N-terminal domain of VIRMA (residues 1–130) is conserved between *Drosophila* (Fig. 2b, c) and humans (Supplementary Fig. 6b, c). We further showed that a region of HAKAI spanning residues 87–105, which is not present in the previously determined structure[40], is critical for this interaction (Supplementary Fig. 6c, lanes 11–12). Similarly, the predicted structured region of the Fl(2)d homologue WTAP (residues 1–249) co-immunoprecipitated with VIRMA (Supplementary Fig. 6d). In case of human VIRMA, we identified two distinct interaction domains. On one hand VIRMA interacts with WTAP through its C-terminal domain (residues 1575–1812) (Supplementary Fig. 6e), as shown for *Drosophila*. We also identified a second WTAP interacting site in the central region of the protein spanning residues 335-1130 (Supplementary Fig. 6e, lane 14). Interestingly, truncating this region further from either side abolished the interaction (Supplementary Fig. 6e, lanes 15–16), suggesting that WTAP binding involves multiple interaction sites or that the truncations disrupt the VIRMA protein fold. Altogether, these results demonstrate that VIRMA serves as an interaction platform for the assembly of HAKAI and WTAP into the MACOM complex in both *Drosophila* and humans (Fig. 2g, h).

**Hakai is required for mRNA m[6]A methylation and alternative splicing of *Sex lethal (Sxl)*.** A key role for m[6]A in *Drosophila* has been shown in alternative splicing of *Sex lethal* (*Sxl*), where it is required for autoregulation in females[41]. In females, Sxl binds to either side of an alternative exon containing a stop codon and blocks the splice sites to skip this exon, which is only included in males. In addition, Sxl also prevents expression of the dosage compensation factor msl-2, which is expressed only in males to upregulate transcription from the single X chromosome twofold. Loss of m[6]A interferes with sexual differentiation in females and increases female lethality due to aberrant dosage compensation[14–16]. Hence, we wondered whether loss of *Hakai* is also required for mRNA m[6]A methylation and interferes with *Drosophila* sex determination and dosage compensation.

We obtained a previously characterized imprecise transposon excision line in the *Hakai* gene[42]. This *Hakai*[1] allele lacks the coding region covering the N-terminus and the RING-finger domain, and is considered to be a null loss of function (Fig. 3a). In addition, we used the CRISPR/Cas9 approach to generate *Hakai*, which encodes an early truncated product (first 57aa). When we crossed *Hakai*[1] to either of the two deficiency alleles (*Df(2L)Exel8041* or *Df(2L)Exel6044*) to normalize genetic background effects, we found that the mutants died in the pupal stage (n = 240). The m[6]A level of *Hakai*[1]/*Df(2L)Exel8041* pupae was reduced compared to the control (Fig. 3b–d), but not completely absent as in *Mettl3*[null] mutants[14].

To test whether Hakai is required for *Sxl* autoregulation, we made use of a genetically sensitized background based on reduced Sxl levels by removal of one copy of *daughterless* (*da*), which is involved in *Sxl* transcription, and one copy of *Sxl* required for *Sxl* autoregulation. In the progeny of a cross between *da*[Df]/+; *Mettl3*[null]/+ females and *Sxl*[7B0] null males, most females died (Fig. 3e). Likewise, also removal of one copy of *Hakai* killed females (Fig. 3e).

Furthermore, when we crossed *Hakai*/*CyO* females, which harbours an early stop codon, to *Df(2L)Exel8041*/*CyO* males to normalize genetic background, we observed strong female lethality compared to *CyO* balancer-carrying control animals (149 females and 144 males, Fig. 3f). Although the two females we obtained did not show sexual transformation, all male flies were flightless (n = 44), as observed for *Mettl3*[null] and *Mettl14*[null] mutants[14–16].

To further confirm the involvement of Hakai in *Sxl* alternative splicing we made use of the female-lethal *vir*[2F] allele[14]. We found that removal of one copy of *Hakai* restored female viability of *vir*[2F]/*Df(2R)BSC778* females by correcting *Sxl* alternative splicing (Fig. 3g), as shown previously for other components of the methyltransferase complex[14,24].

Lastly, to demonstrate a role for Hakai in the sexual differentiation of *Drosophila* females we made use of the *vir*[2F]/*vir*[ts] genetically sensitized background, which occasionally shows patches of darkly pigmented male tissue in abdominal segments 5 and 6 in these females (Fig. 3j–l). Strikingly, removal of one copy of *Hakai* led to male pigmentation in the vast majority of these females (Fig. 3h, m) and a switch of *Sxl* alternative splicing to the male mode, including the otherwise skipped male exon (Fig. 3i, lane 5). Some females also displayed intersexual development of genitals (Fig. 3n). Intriguingly, loss of *Hakai* in the *vir*[2F]/*vir*[ts] genetically sensitized background revealed tissue-specificity in *Sxl* regulation as alternative splicing in the front part of females (head and thorax) was not altered, and also these females did not display male sex combs (Fig. 3h), suggesting the m[6]A pathways main function could be to guarantee robust *Sxl* alternative splicing across different tissues.

**Hakai regulates the m[6]A pathway in *Drosophila*.** To further corroborate Hakai as a genuine m[6]A writer in *Drosophila*, we

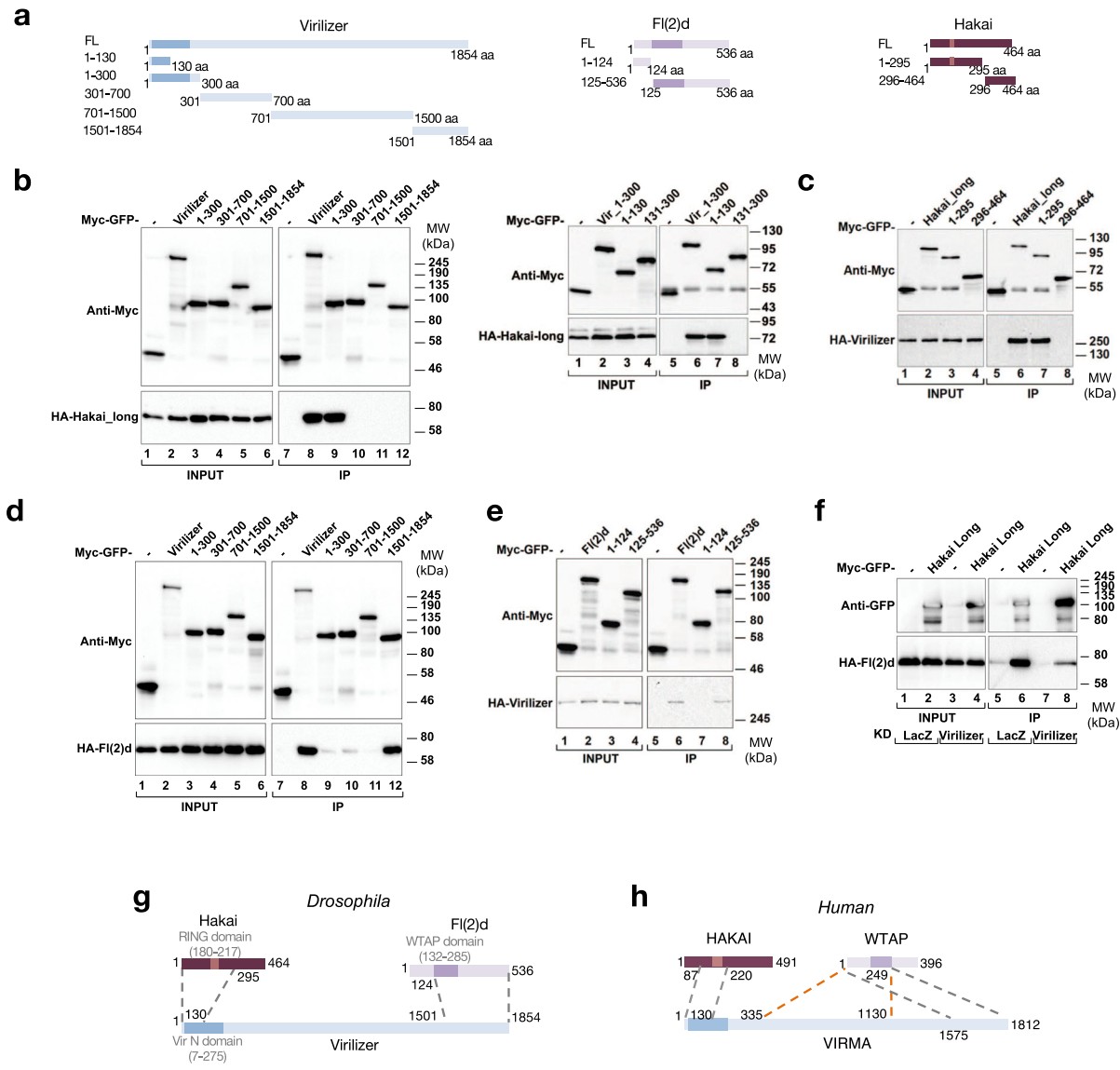

**Fig. 2 Virilizer acts as a scaffold between Hakai and Fl(2)d. a** Schematic representation of proteins and protein fragments used for *Drosophila* co-immunoprecipitation assays. **b–e** Co-immunoprecipitation experiments were carried out with lysates prepared from S2R+ cells transfected with Myc-GFP-tagged Virilizer (full length or fragments) and HA-tagged Hakai-long (**b**), Myc-GFP-tagged Hakai-long (full length or fragments) and HA-tagged Virilizer (**c**), Myc-GFP-tagged Virilizer (full length or fragments) and HA-tagged Fl(2)d (**d**), Myc-GFP-tagged Fl(2)d (full length or fragments) and HA-tagged Virilizer (**e**). In control lanes, S2R+ cells were transfected with Myc-GFP alone and an identical HA-containing protein. Extracts were incubated with magnetic agarose GFP binder beads and immunoblotted using anti-Myc and anti-HA antibodies (**b–e**), as indicated. Two percent of input was loaded. The experiment was performed in the presence of RNase A. Images shown are representative of two biological replicates. **f** Co-immunoprecipitation experiments were carried out with lysates prepared from S2R+ cells transfected with Myc-GFP-tagged Hakai-long and HA-tagged Fl(2)d upon control (*LacZ*) or *vir* KDs. In control lanes, S2R+ cells were transfected with Myc-GFP alone and an identical HA-containing protein. Extracts were incubated with magnetic agarose GFP binder beads and immunoblotted using anti-GFP and anti-HA antibodies. Two percent of input was loaded. **g**, **h** Schematic representing the interaction domains between Hakai, Vir and Fl(2)d derived from co-IP experiments in *Drosophila* (**g**) or HEK393T cells (**h**). Orange dotted lines indicate the second interaction domain between Human WTAP and VIRMA. Source data for western blots are provided as a Source Data file.

depleted its product in S2R+ cells and compared its effect with the KD of other m6A pathway components. In agreement with Hakai being part of the m6A writer complex, its loss led to a significant reduction of m6A levels on mRNA as measured by mass spectrometry (Fig. 4a). However, this reduction (32%) was not as pronounced as the m6A reduction observed upon depletion of the Mettl3/Mettl14 heterodimer (59%), which is consistent with the results obtained by TLC from *Hakai*[1]/*Df* mutant pupae (Fig. 3c, d). We next investigated its involvement in the regulation of m6A-dependent splicing events. We previously reported

changes in the splicing pattern of several transcripts upon KD of individual m6A writers[15,24]. Splicing isoform quantification by RT-qPCR of two of the affected transcripts, *fl(2)d* and *Hairless*, resulted in a comparable isoform shift in *Hakai* KD and *Mettl3/ Mettl14* KD in S2R+ cells (Fig. 4b). On a transcriptome-wide level, depletion of *Hakai* resulted in changes in gene expression and splicing that substantially overlap with changes occurring upon KD of other components of the m6A machinery (Fig. 4c; Supplementary Fig. 7). In particular, *Hakai* KD led to an increase of both alternative 5′ splice site usage and intron retention,

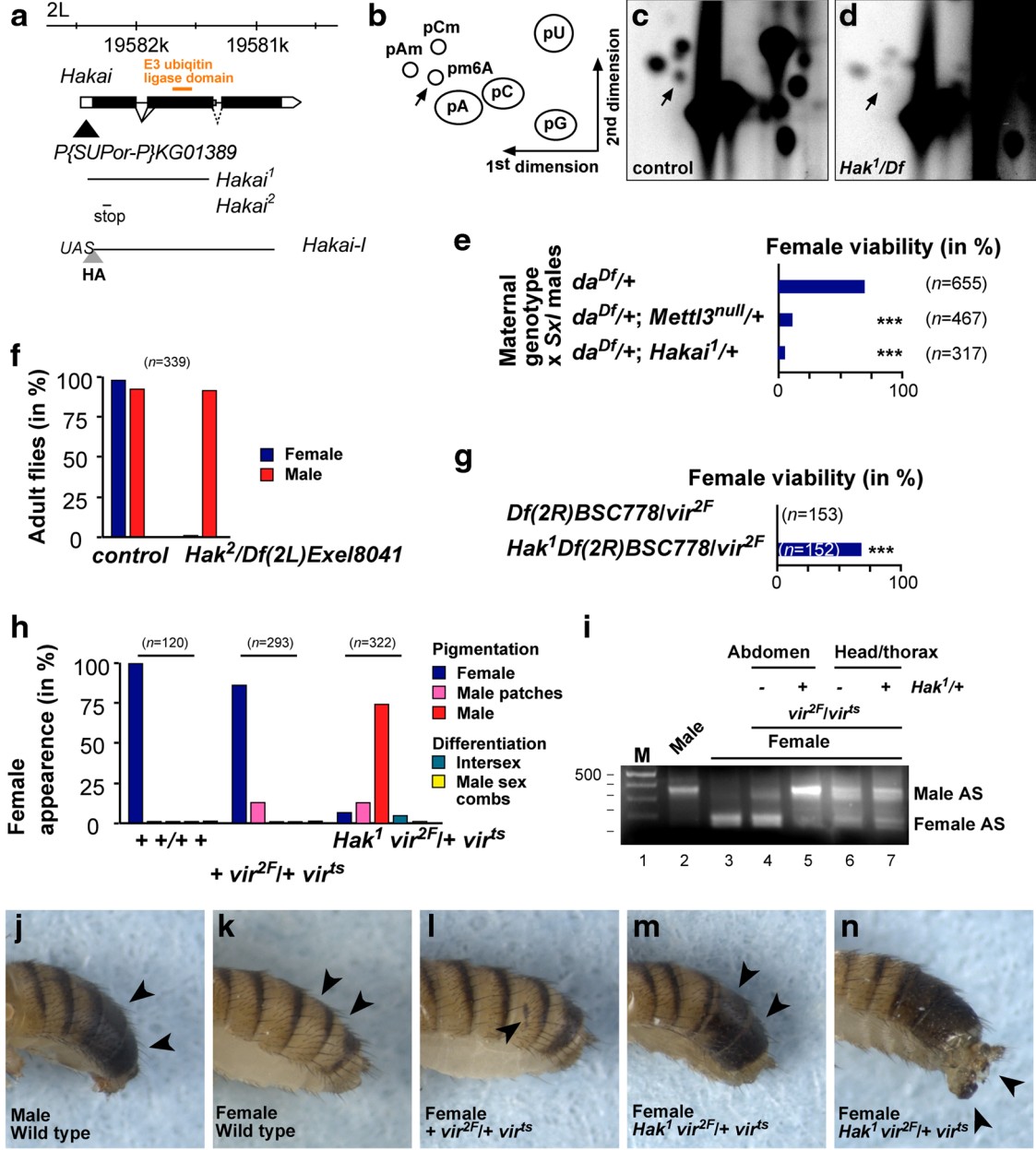

**Fig. 3 Hakai is required for m$^6$A methylation and sex determination by regulating *Sex-lethal* alternative splicing. a** Schematic of the *Hakai* genomic locus depicting the transposon (black triangle) used to generate the deletion in the *Hakai*[1] allele and the premature stop codon present in the *Hakai* allele, and the epitope-tagged UAS constructs of the short and long isoforms. Schematic diagram of a 2D thin-layer chromatography (TLC) depicting standard and methylated nucleotides (**b**), and TLCs depicting m$^6$A in control (**c**) and *Hakai*[1]/*Df(2L)Exel8041* pupae (**d**). **e** Viability of female flies from a cross of the indicated genotypes mated with *Sxl*[7BO] males. The loss of one copy of *Mettl3* or *Hakai* significantly reduces female survival in a genetic background where one copy of *Sxl* and *da* are absent. Viability was calculated from the numbers of females compared with males, and statistical significance was determined by a $\chi^2$ test (Graphpad Prism). ***$P \leq 0.0001$. Unpaired two-tailed Student's *t*-test for unequal variances. **f** Viability of *Hakai/Df(2L)Exel8041* flies. **g** The viability of female flies with homozygous *vir*[2F] mutation can be rescued by the loss of a single copy of *Hakai*. Viability was calculated from the numbers of homozygous *vir*[2F] females compared with heterozygous balancer-carrying siblings, and statistical significance was determined by a $\chi^2$ test (Graphpad Prism). ***$P \leq 0.0001$. **h–n** External sexual differentiation (**h**, **j–n**) and *Sxl* alternative splicing in abdomen and head/thorax fraction (**i**) of control, *vir*[2F]/*vir*[ts] and *Hakai*[1] *vir*[2F]/*vir*[ts] female flies. The gel shown in (**i**) is a representative of two biological replicates. The marker is a 100 bp DNA ladder with 500 bp indicated on top. Note that *Sxl* alternative splicing in *Hakai*[1] *vir*[2F]/*vir*[ts] female abdomens is switched to the male mode and that these females display male pigmentation (**h**, **m**), but no male sex combs (**h**). Source data for TLC, fly numbers and RT-PCR gels are provided as a Source Data file.

consistent with our previous findings on the individual MACOM component depletion (Fig. 4d[24]). Note that depletion of MACOM components has a stronger impact on gene expression and splicing compared to the loss of MAC components. This is consistent with previous genetic data showing that Mettl3 and Mettl14 are dispensable for fly viability while MACOM subunits are not[14–16], supporting additional function(s) for MACOM components. Altogether, these results demonstrate that Hakai is a bona fide component of MACOM and is required for m$^6$A biogenesis and function.

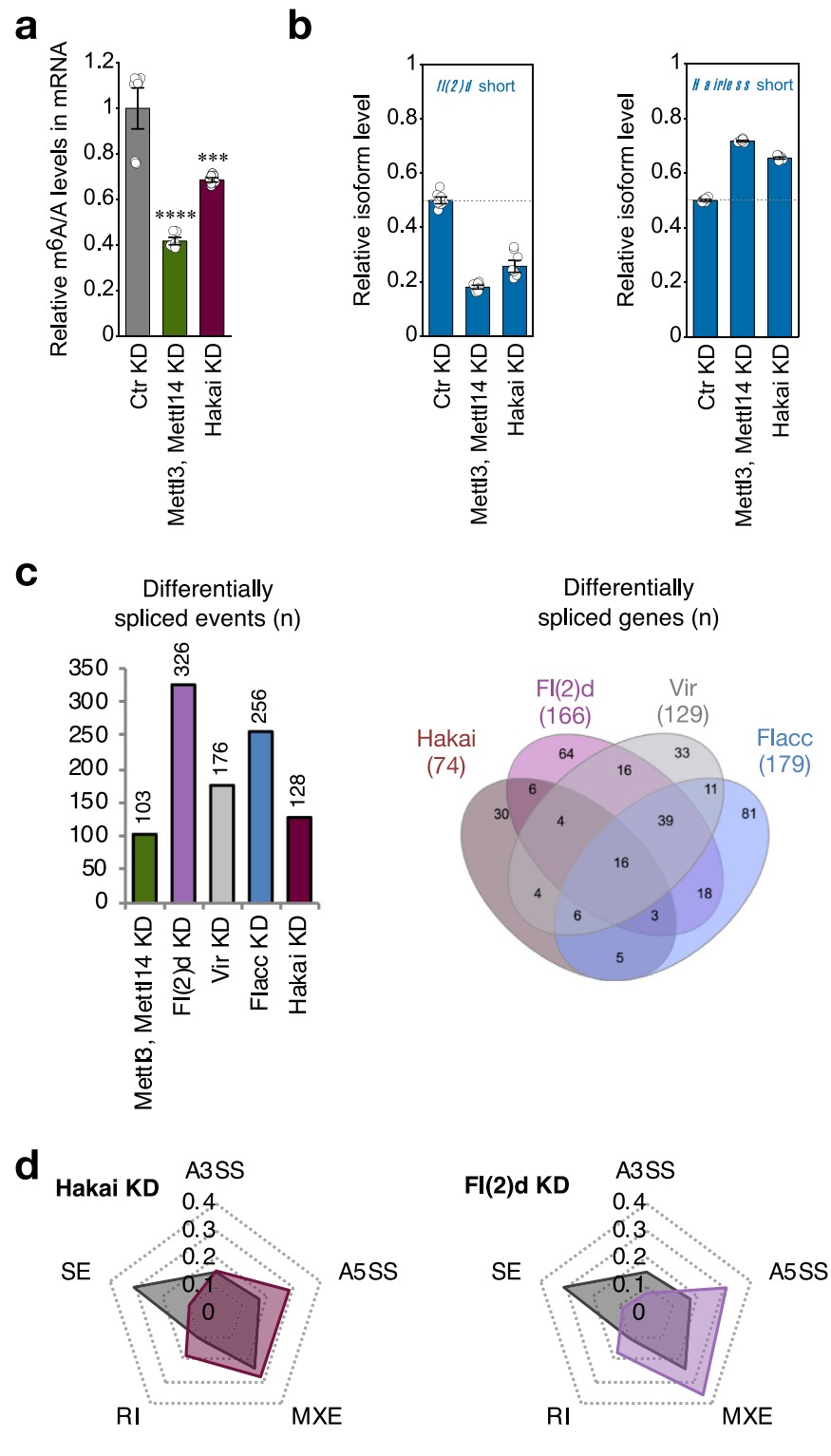

**Fig. 4 Hakai regulates m⁶A levels and m⁶A-dependent splicing events. a** LC–MS/MS quantification of m⁶A levels in either control samples or mRNA extracts depleted for the indicated proteins in S2R+ cells. The bar chart shows the mean with standard error (SE) of three biological replicates and three technical measurements. KD of *Hakai* results in substantial reduction of m⁶A levels. ***$P = 7.49E-04$ (Hakai KD), ****$P = 7.51E-06$ (Mettl3, Mettl14 KD). Unpaired two-tailed Student's *t*-test for unequal variances. **b** Relative isoform quantification of m⁶A-regulated genes (*fl(2)d* and *Hairless*) upon depletion of the indicated components. The bar chart shows the mean with standard error (SE) of three biological replicates and three technical measurements. Hakai is required for m⁶A-dependent splicing regulation. **c** Number of differentially spliced events upon knockdown of the indicated proteins (left) and common differentially spliced targets (right) (FDR < 0.1). **d** Radar charts display relative distribution of differentially spliced events upon knockdown of Fl(2)d and Hakai. Alternative 5'splice site (A5SS) selection and intron retention (RI) are overrepresented events upon loss of m⁶A writer components. Source data for m⁶A measurement, qPCR and RNA-seq are provided as a Source Data file.

**The Hakai ubiquitination domain but not its activity is required for maintaining MACOM integrity**. We next sought to address the molecular role of Hakai within MACOM. As Hakai is a well-studied E3 ubiquitin ligase in mammals, we wondered whether its ubiquitination activity plays any role within the $m^6A$ pathway. To address this possibility, we generated constructs expressing either the wild-type *Hakai* cDNA (*Hakai^WT*) or a cDNA containing point mutations in the RING domain (*Hakai^ΔRING*) that should abolish ubiquitination activity (see "Methods"). We then performed rescue experiments in our *Hakai* mutant flies and found that while the wild-type form was able to rescue the lethality, the mutated version failed to do so. This indicates that the RING domain is required for fly viability.

We therefore wondered if Hakai might regulate $m^6A$ levels by ubiquitination of MACOM components. To investigate this possibility we examined our previous ubiquitylome datasets from *Drosophila* S2 cells and found that Fl(2)d and Nito were among the ubiquitinated proteins[43]. Thus, we cloned both proteins in a GFP-tag containing vector and expressed them in control and Hakai depleted S2R+ cells to monitor their ubiquitination. We immunoprecipitated both proteins under stringent 8 M Urea conditions and while we could not detect any ubiquitination signal for Nito, Fl(2)d appeared to be polyubiquitinated, as shown by a strong shift in molecular weight by more than 100 kDa (Supplementary Fig. 8a). Using mass spectrometry analysis, we could map two of the four previously identified sites (K236 and K245) in Fl(2)d (Supplementary Data 2), residing in the region required for homodimerization and interaction with Vir (Fig. 2e, Supplementary Fig. 4a). We noticed that Fl(2)d ubiquitination was reduced in the *Hakai* KD condition; however, the overall level of immunoprecipitated Fl(2)d was also diminished. We therefore repeated this experiment in control condition and after proteasome inhibition to prevent protein degradation. This experiment confirmed our previous observations; Fl(2)d was ubiquitinated and its protein intensity was strongly reduced upon *Hakai* KD. However, upon proteasome inhibition, levels of Fl(2)d as well as its ubiquitination remained unchanged despite efficient *Hakai* depletion (Supplementary Fig. 8b). This indicates that Hakai is not responsible for Fl(2)d ubiquitination, but is required for its stability.

To get better insight into the function of Hakai as an E3 ubiquitin ligase and find other putative targets, we next performed a ubiquitylome analysis in S2R+ cells in control versus *Hakai* KD condition. Cells isotopically labelled with heavy amino acids were depleted for *Hakai* and cells isotopically labelled with light amino acids served as a control in the forward experiment. A vice versa depletion was performed in the reverse experiment (Supplementary Fig. 8c). Proteins were digested with endo-proteinase Lys-C and peptides were further enriched with di-glycine-lysine remnant-recognizing antibody to identify ubiquitination sites via LC-MS/MS. We found over 3000 ubiquitination sites, but unexpectedly not a single site was reproducibly reduced in response to *Hakai* depletion and only one site in SesB was 1.5-fold increased (Fig. 5a, Supplementary Data 3). Therefore, this experiment suggests that Hakai does not act as an E3 ubiquitin ligase in *Drosophila* S2R+ cells. Alternatively, it is possible that its ubiquitination activity depends on specific external stimuli or that we have not quantified the ubiquitination sites that are regulated by Hakai due to the limited depth of the analysis.

We wondered what could explain the apparent discrepancy between our in vivo results, indicating the importance of the Hakai RING domain and the data obtained from S2R+ cells. Previous crystal structure in mammals showed that the RING domain is required for HAKAI dimerization[40]. One possibility could be that the point mutations we generated in the RING

domain alter the ability of Hakai to dimerize and perhaps to interact with other MACOM components. To test this idea, we performed co-immunoprecipitation experiments by co-transfecting S2R+ cells with Myc-GFP-tagged Hakai either wild type or mutated in the RING domain and other components of MACOM, including Hakai itself. While WT Myc-GFP-Hakai strongly immunoprecipitated HA-Hakai, the RING mutant failed to do so, indicating that, like in mammals, the RING domain is required for Hakai homodimerization (Fig. 5b). More importantly, interactions with Fl(2)d and Vir were also strongly compromised (Fig. 5c, d). We interpret this result as an indication that Hakai dimerization is likely required for its association with other MACOM components.

**Hakai is required for stabilization of MACOM components**. Our results so far indicate that Hakai does not possess any ubiquitin activity towards MACOM components, and that it also does not serve as a scaffold to permit the assembly of these components. Nevertheless, as mentioned above, we found that *Hakai* depletion strongly destabilized Fl(2)d protein levels. In addition, we noticed in our co-immunoprecipitation experiments that the level of protein input for Fl(2)d and Vir was consistently reduced upon *Hakai* KD (Supplementary Fig. 5c, d), suggesting that Hakai may be required to stabilize these subunits. For this reason, we had to transfect twice the amount of these components to have comparable input level and therefore interpretable co-immunoprecipitation data. To confirm these observations, we took an unbiased approach by performing a global proteome analysis in S2R+ cells following *Hakai* depletion. As expected, Hakai levels were strongly reduced, indicating a successful KD (Fig. 6a). More strikingly, we identified seven additional proteins that were strongly down regulated, and among them three MACOM components: Fl(2)d, Flacc and Vir (Supplementary Data 4). The extent of downregulation for Vir was similar as for Hakai, indicating that Hakai loss has a particularly strong impact on Vir stability. In contrast, we did not detect any substantial changes in the protein levels of Nito, Mettl3 and Mettl14. We could confirm by western blot the specific reduction of the Fl(2)d level (Fig. 6b, c and Supplementary Fig. 9d). However, immunofluorescence assay showed that its nuclear localization was not affected (Supplementary Fig. 8d).

To examine whether the loss of Fl(2)d or Vir also impacts the stability of other $m^6A$ writer components, we repeated the S2R+ proteome analysis after depletion of these proteins. We found that Fl(2)d protein levels were significantly reduced upon *vir* KD and vice versa (Supplementary Fig. 9a-c). Similarly, as shown upon *Hakai* depletion, Flacc level was also decreased. However, in both cases, levels of Hakai and Nito were not affected. This was also confirmed by western blot (Fig. 6d and Supplementary Fig. 9e). Collectively, these results indicate that Hakai functions in stabilizing three MACOM components and this is likely the reason why $m^6A$ levels are reduced upon its depletion.

**The role of Hakai in MACOM component stabilization is conserved in human**. Given the high conservation of Hakai with its human ortholog and its interaction with Vir, we wondered if HAKAI function within MACOM is conserved in humans. To this end, we depleted human *HAKAI* and *VIRMA* in HeLa and U2OS cells and monitored the protein levels. We observed that the loss of *VIRMA* had no effect on HAKAI level (Fig. 6e), as also shown with the *Drosophila* homologues. In contrast, depletion of *HAKAI* strongly reduced VIRMA levels but not the levels of RBM15, the human ortholog of Nito. Altogether, these findings indicate that the role of HAKAI within MACOM is conserved in humans and that Hakai is indispensable for maintaining the

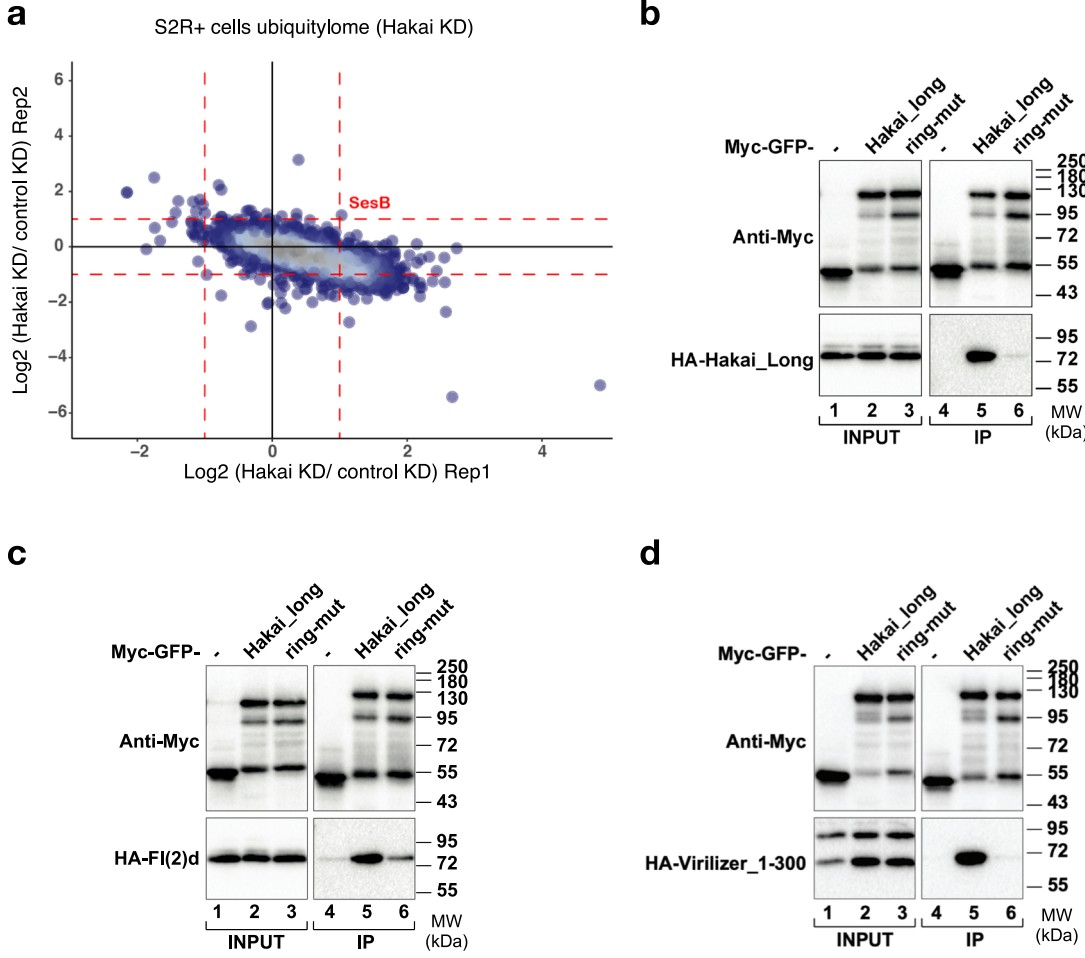

**Fig. 5 Hakai RING domain is required for interaction with MACOM components. a** Mass spectrometry analysis of Hakai-dependent ubiquitinated proteins in S2R+ cells. Scatter plot of normalized forward versus inverted reverse experiments plotted on a log2 scale. The threshold was set to a twofold enrichment or depletion (red dashed line). One protein in the top right quadrant is enriched in both replicates. Hakai depletion does not affect global ubiquitination levels in *D. melanogaster* S2R+ cells. A complete list of quantified ubiquitylation sites after HAKAI depletion can be found in Supplementary Data 3. **b**–**d** Co-immunoprecipitation experiments were carried out with lysates prepared from S2R+ cells transfected with Myc-GFP-tagged Hakai-long (WT or RING mutant) and HA-tagged Hakai-long (**b**), HA-tagged Fl(2)d (**c**) and HA-tagged Virilizer fragment (**d**). In control lanes, S2R+ cells were transfected with Myc-GFP alone and an identical HA-containing protein. Extracts were incubated with magnetic GFP binder beads and immunoblotted using anti-Myc and anti-HA antibodies, as indicated. Two percent of input was loaded. The experiment was performed in the presence of RNase A. Images shown are representative of two biological replicates. Source data for Western blots are provided as a Source Data file.

functionality of m⁶A writer by ensuring the stability of MACOM components (Fig. 7).

## Discussion

We recently showed that two conserved sub-complexes, MAC and MACOM interact to deposit m⁶A on mRNA in flies and mice[24]. While the structure of the catalytic MAC, which consists of the heterodimer METTL3 and METTL14, has been thoroughly characterized[28–30], our knowledge of MACOM is limited. In particular, the full composition, assembly and exact function of each subunit have remained unclear. Our study identifies Hakai as an integral component of MACOM in *Drosophila* and human cells. In line with this function, we show that Hakai interacts with Vir and other MACOM components, and its depletion reduced m⁶A levels and led to altered gene expression, resembling loss of other MACOM subunits. Furthermore, flies lacking *Hakai* are lethal and display aberrant splicing of *Sex lethal*, consistent with the role of MACOM in sex determination and dosage

compensation pathways. The few individuals that escape lethality are flightless, as shown earlier in other mutants of the m⁶A pathway. Mechanistically, we found that Hakai is required to stabilize several MACOM components, likely explaining its requirement for m⁶A deposition.

The question that arises is whether all MACOM components have now been identified. We and others have validated five factors, which include Fl(2)d, Vir, Flacc, Nito and Hakai (this study and[14–16,24,33]). Earlier biochemical studies estimated a molecular weight of 875 kDa for the large form of the human methyltransferase complex[44]. The calculated total molecular weight of the combined five factors corresponds to 600 kDa, which suggests that the complex contains additional factors or multiple copies of the known factors. Our data show that Hakai, Fl(2)d and Nito have the ability to self-interact (Supplementary Fig. 4). If we assume that these three factors are present as dimers, the total weight reaches up to 868 kDa, which would be consistent with the predicted mass of MACOM. Nevertheless, additional

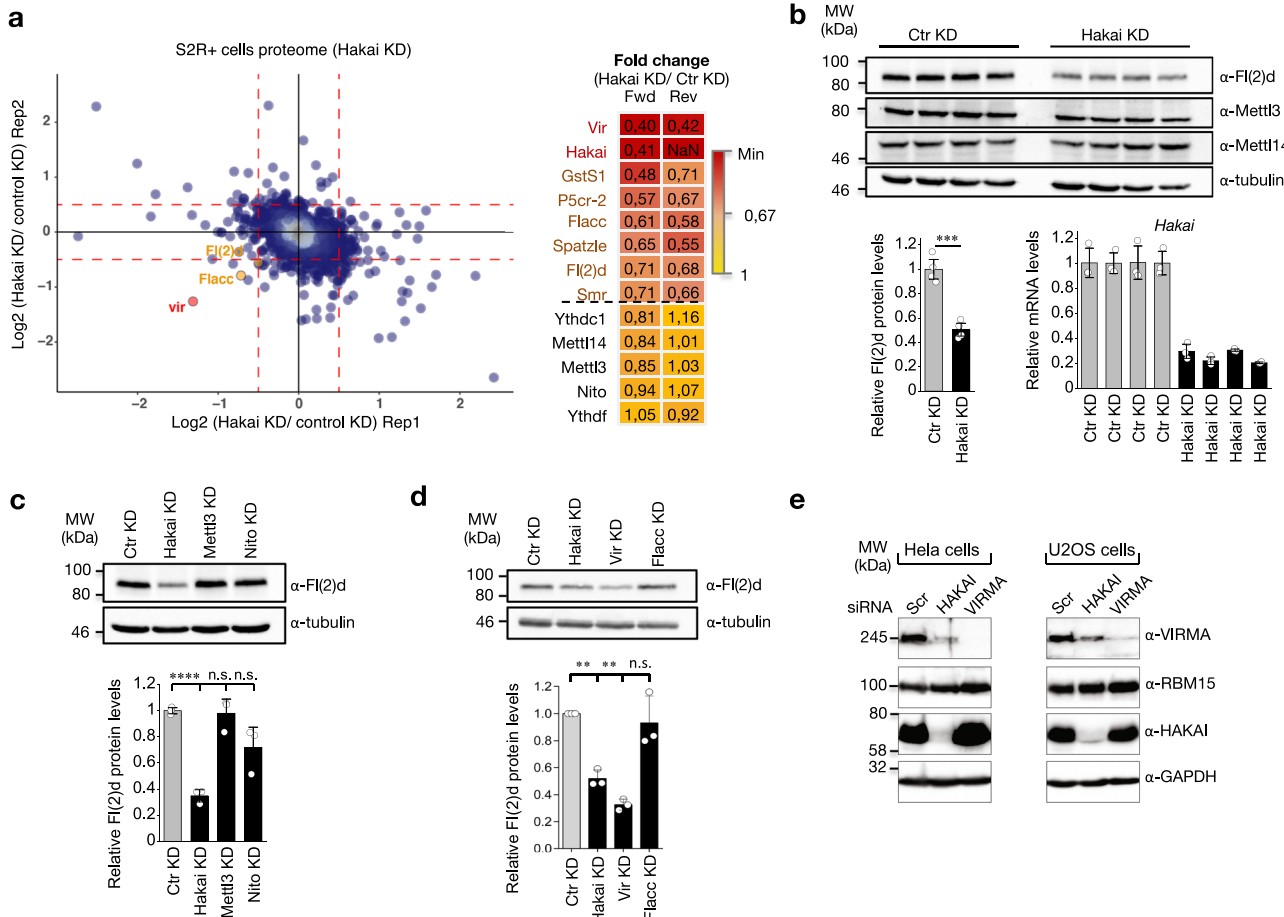

**Fig. 6 Hakai regulates the stability of other MACOM components. a** Mass spectrometry analysis of Hakai-dependent proteome in S2R+ cells. Scatter plot of normalized forward versus inverted reverse experiments plotted on a log2 scale. The threshold was set to a 1.4-fold enrichment or depletion (red dashed line). Proteins in the bottom left quadrant are decreased in both replicates. Heat map of proteins whose levels were reduced by >1.4-fold in both replicates of the whole proteome analysis. Other components of the m6A writer complex and reader proteins are shown for comparison. A complete list of quantified proteins can be found in Supplementary Data 4. **b** Levels of endogenous Fl(2)d, Mettl3 and Mettl14 proteins from control cells and cells depleted for Hakai were analyzed by western blot. Tubulin was used as a loading control. Quantification of Fl(2)d, Mettl3 and Mettl14 levels from blots shown below (left). The bar chart shows the mean with standard error (SE) of four biological replicates. ***$P = 2.48E-04$ (Hakai KD). Unpaired two-tailed Student's t-test for equal variances. Hakai depletion strongly destabilizes Fl(2)d, but not Mettl3 or Mettl14 proteins. Relative expression levels of *Hakai* are shown as a validation of its KD efficiency. The bar chart shows the mean with standard error (SE) of three technical measurements (Bottom right). **c, d** Analysis of Fl(2)d levels upon *Hakai*, *Mettl3*, *nito* (**c**) or *Hakai*, *vir*, *Flacc* (**d**) depletion. Protein lysates from control and depleted cells were analyzed by western blot for levels of endogenous Fl(2)d protein. Tubulin was used as a loading control. One representative experiment is shown and quantification of three biological replicates is shown below. The bar chart shows the mean with standard error (SE). (**c**) ****$P = 4.74E-05$ (Hakai KD), n.s.$P = 0.80029$ (Mettl3 KD) and n.s.$P = 0.05398$ (Nito KD). (**d**) **$P = 0.0057$ (Hakai KD), $P = 0.0012$ (Vir KD) (Hakai KD), and n.s.$P = 0.6061$ (Flacc KD). Unpaired two-tailed Student's t-test for equal variances. **e** Western blots were carried out with lysates prepared from HeLa and U2OS cells transfected with scrambled siRNA or siRNA against HAKAI or VIRMA. Extracts were immunoblotted using the indicated antibodies. Depletion of HAKAI reduced VIRMA levels while depletion of VIRMA had no impact on HAKAI levels. Images shown are representative of two biological replicates. Source data for western blots, measurement of protein levels and qPCR are provided as a Source Data file.

biochemical and structural characterization will be required to confirm the exact identity and stoichiometry of the different complex components.

While Hakai is undoubtedly a core component of the complex, it is surprising that its gene inactivation results in milder phenotypes in comparison to the inactivation of other MACOM components. Indeed, the few females that escaped developmental lethality in the *Hakai* loss-of-function mutant did not display male sex combs and this phenotype was also not observed in a sensitized background with reduced *Hakai* dosage. However, the females did show male pigmentation on their abdomen, indicating tissue-specific alteration of sex determination. These milder defects are consistent with our quantification of the m6A level in the *Hakai* mutant (Fig. 3), which appeared reduced but

not completely absent, as observed in the *Mettl3* mutant[14]. Also, *Sxl* splicing showed tissue-specific alterations, indicative of local requirement for this factor. These results are consistent with a previous study in *Arabidopsis* showing less pronounced impact of Hakai on the m6A levels as well as on organismal development[26]. This apparent discrepancy may be explained by the function we uncovered in this work. Our data show that upon depletion of *Hakai* the level of some of the other MACOM components is reduced but not completely lost (Fig. 6), which suggest that the remaining MACOM could still support methylation. In this case, tissue-specific requirement may be mediated through differential expression of factors that may impact on this residual interaction and could determine tissue-specific levels of m6A. Additional work would be required to test this hypothesis.

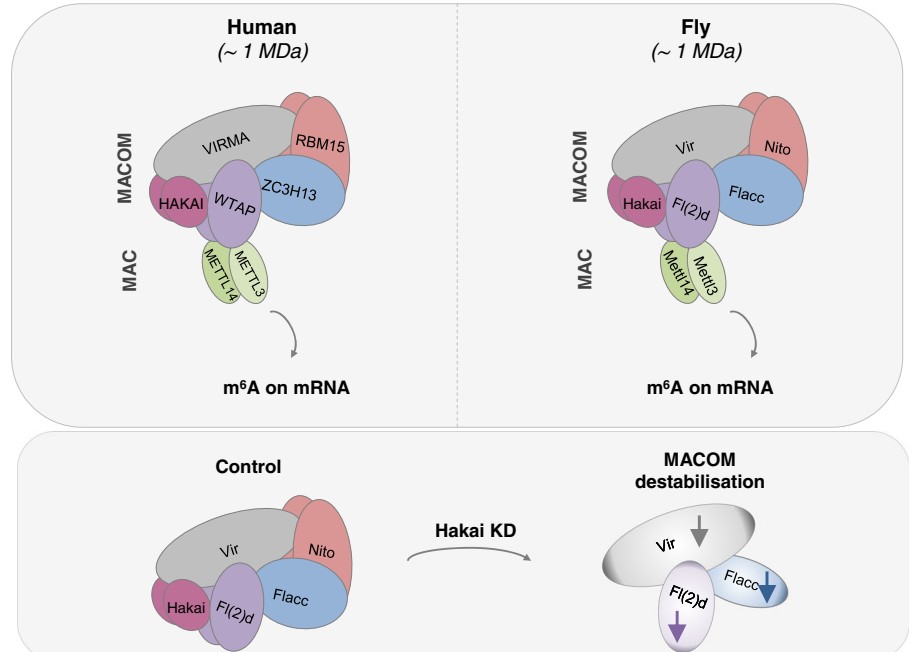

**Fig. 7 Model showing the composition of the m⁶A methyltransferase complex in fly and human and the impact of Hakai on MACOM integrity.** Top left and right represent the human and fly m⁶A methyltransferase complexes, respectively. (Bottom) The depletion of Hakai leads to a reduction in protein levels for Vir, Flacc and Fl(2)d.

In vertebrates, Hakai was shown to ubiquitinate E-cadherin and promote its degradation[35]. In contrast, our data from *Drosophila* cells provide no evidence for ubiquitination activity towards MACOM components or any other proteins. Instead, our work strongly suggest that Hakai is required for the stability of three MACOM components, Vir, Fl(2)d and Flacc, independently of its enzymatic activity. The question that remains is how does Hakai exert this function. It was previously shown that the so-called "orphan proteins" are unstable and get degraded if their protein partners that constitute a common complex are absent. Destabilization can be triggered due to aberrant protein folding, altered localization or because of exposure of normally protected protein binding interfaces[45]. It is therefore possible that Hakai stabilizes other MACOM components via one of these mechanisms. If this model is true, other components of the complex are also expected to stabilize each other. Indeed, we found that protein levels of Vir, Fl(2)d and Flacc were also strongly reduced upon loss of Vir or Fl(2)d (Fig. 6). Interestingly, in mES cells a strong reduction of Wtap levels was previously reported upon loss of Virma[27] and a strong destablization of Zc3h13 was observed upon depletion of Virma, Wtap or Hakai[34], suggesting a conserved mechanism for Hakai-Vir-Fl(2)d-Flacc stabilization between flies and mice. We further demonstrated that protein levels of Hakai and Nito are unperturbed upon depletion of other MACOM components, suggesting that Hakai and Nito might function in additional processes not linked to MACOM or m⁶A-deposition. This is in agreement with observations that Hakai also localizes in the cytoplasm, and with a study that found Nito (RBM15) in an evolutionary conserved protein complex with proteins unrelated to remaining MACOM components[39].

Given our insights from the current study, we propose the following model of MACOM assembly: (i) Fl(2)d-Vir-Hakai form a minimal protein unit that is required for the assembly and functionality of the remaining methyltransferase complex. (ii) Stability of Fl(2)d and Vir depends on each other and Hakai, but is largely independent of Flacc and Nito. (iii) Fl(2)d-Vir-Hakai can interact with MAC, however, this is not sufficient for m⁶A

deposition. (iv) Joining of Flacc and Nito is essential for the formation of a complete MACOM complex that can bind and methylate its targets together with MAC.

In conclusion, our work revealed the requirement of Hakai for m⁶A deposition in *Drosophila*. The apparent dependency of Fl(2)d, Vir and Flacc for each other's stability might be an important mechanism that maintains an equilibrium of protein stoichiometry for a complete complex assembly in order to prevent aberrant interactions of orphan subunits and unwanted m⁶A installation. Given the critical role of m⁶A in multiple physiological processes, it will be important to address whether perturbation of this equilibrium, for instance, by single nucleotide polymorphism, may impact the development or severity of pathological conditions.

## Methods

***Drosophila*** **stocks and genetics**. *Drosophila melanogaster* CantonS was used as the wild-type control. All *Hakai* alleles and corresponding chromosomal deficiencies (*Df(2L)Exel8041* and *Df(2L)Exel6044*) were re-balanced with a *CyO* balancer that is marked by *actinGFP* and *w+* to determine the lethal stage and collect homozygous *Hakai* null animals. Sexing at the larval stage was done by monitoring the presence of the male gonad.

The *P{VSH330548}attP40 Hakai* RNAi line was obtained from Vienna Drosophila Resource Center). *Hakai* was generated using the CRISPR-Cas9 system following the previously described procedure[46]. Two independent guide RNAs were designed using the gRNA design tool: www.crisprflydesign.org (Supplementary Data 5). Oligonucleotides were annealed and cloned into pBFv-U6.2 vector (National Institute of Genetics, Japan). The vector was injected into embryos of TBX-0002 (y1, v1, P{nos-phiC31\int.NLS}X; attP40 (II)) flies by Bestgene Inc. Flies were crossed with TBX-0008 (y2 cho2 v1/Yhs-hid; Sp/CyO) flies to identify positive recombinant flies by eye colour marker (Vermillon). Males were further crossed with CAS-0001 (y2, cho2, v1; attP40(nos-Cas9)/CyO) females. Males carrying *nos*-Cas9 and U6-gRNA transgenes were screened for the expected deletion using oligos in Supplementary Data 5.

For genetic interaction studies, we used *Mettl3null* and *vir* alleles as described[24]. The *da* deficiency was *Df(2L)BSC209*.

**Cloning**. The plasmids used for immunohistochemistry and co-immunoprecipitation assays (shown in Fig. 1d, e) were constructed by cloning the corresponding cDNA in the pAC vector with the N-terminal Myc- tag (Hakai short and Hakai long) or N-terminal Myc-GFP tag (Fl(2)d and Nito)[15]. For plasmids

with the C-terminal HA tag, the corresponding cDNA was cloned in the Gateway-based vector (pPWH) obtained from Drosophila Genomics Resource Center (DGRC) at Indiana University as described in ref. [15]. To generate the RING mutant construct the corresponding mutations were introduced: histidine at position 183 was replaced by alanine in combination with cysteine to alanine conversion for the residues 165, 168, 181, 186, 189, 198 and 201. The stability of the protein was not affected by these mutations. Hakai long and Ring mutant cDNA were cloned in the pUAST-eGFP-attB vector between BamH1 and Xba1 sites and the constructs were sent to BestGene for injections.

Myc-GFP construct was generated by cloning Myc tag in directional cloning within the Kpn1 restriction site in pAc5.1B-EGFP-V5-His vector. This construct also has V5-His in frame after EGFP. This construct was used as negative control for the experiment. Then all cDNAs were cloned in this vector for the subsequent experiments. For HA-tagged constructs all the cDNAs were cloned in pAc5.1B-lambdaN-HA vector in suitable restriction sites.

**Cell culture, RNA interference and transfection**. Drosophila S2R+ cells were grown in Schneider's medium (Gibco) supplemented with 10% FBS (Sigma) and 1% penicillin–streptomycin (Sigma). For RNA interference (RNAi) experiments, PCR templates for the dsRNA were prepared using T7 megascript Kit (NEB). dsRNA against bacterial β-galactosidase gene (lacZ) was used as a control for all RNA interference (RNAi) experiments. S2R+ cells were seeded at the density of $10^6$ cells/ml in serum-free medium and 7.5 µg of dsRNA was added to $10^6$ cells. After 6 h of cell starvation, serum supplemented medium was added to the cells. dsRNA treatment was repeated after 48 and 96 h and cells were collected 24 h after the last treatment. Effectene (Qiagen) was used to transfect vector constructs in all overexpression experiments following the manufacturer's protocol.

For knockdown in HeLa and U2OS cells, siRNAs against VIRMA (catalogue no. s24832), HAKAI (catalogue no. s36537) as well as negative control siRNA were purchased from Ambion. siRNAs were transfected with Lipofectamine 2000 RNAi MAX (Invitrogen) according to the manufacturer's protocol.

**Drosophila cell lines**. Drosophila S2R+ cells were embryonic-derived cells obtained from the DGRC (at Indiana University; FlyBase accession FBtc0000150). Drosophila BG3 cells were derived from larval nervous system and also obtained from the DGRC(at Indiana University; FlyBase accession FBtc0000068). Both cell lines were tested for Mycoplasma infection by RNA-seq experiments.

**Drosophila staging**. The staging experiment was performed as described previously[15] using D. melanogaster w1118 flies. A total of three independent samples was collected for each Drosophila stage as well as for heads and ovaries. Samples from the staging experiment were used for RNA extraction to analyze m6A abundance in mRNA and expression levels of different transcripts during Drosophila development.

**RNA isolation and mRNA purification**. Total RNA from S2R+ cells was isolated using Trizol reagent (Invitrogen), and DNA was removed with DNase I treatment (New England Biolabs). Fly heads from 3- to 5-day-old flies were separated and homogenized in Trizol prior to RNA isolation. mRNA was isolated by two rounds of purification with Dynabeads Oligo d(T)25 (New England Biolabs).

**RT-PCR**. cDNA was prepared using M-MLV reverse transcriptase (Promega). Transcript levels were quantified using Power SYBR Green PCR master mix (Invitrogen) and the oligonucleotides indicated in Supplementary Data 5.

For Sxl quantification, Total RNA was extracted using Tri-reagent (SIGMA) and reverse transcription was done with Superscript II (Invitrogen) according to the manufacturer's instructions using an oligodT17V primer. PCR for Sxl was done for 40 cycles with 1 µl of cDNA with primers Sxl F2 (ATGTACGGCAACAATAATCCGGGTAG) and Sxl R2 (CATTGTAACCACGACGCGACGATG). Experiments included three biological replicates.

**LC–MS/MS analysis of m6A levels**. mRNA samples for LC-MS/MS analysis were prepared as mentioned above. Three hundred nanograms of purified mRNA was digested with 0.3 U Nuclease P1 from Penicillium citrinum (Sigma-Aldrich, Steinheim, Germany) and 0.1 U Snake venom phosphodiesterase from Crotalus adamanteus (Worthington, Lakewood, USA) in 25 mM ammonium acetate, pH 5, supplemented with 20 µM zinc chloride for 2 h at 37 °C. Remaining phosphates were removed by 1 U FastAP (Thermo Scientific, St Leon-Roth, Germany) in the manufacturer-supplied buffer, in a 1 h incubation at 37 °C. The resulting nucleoside mix was then spiked with 13C stable isotope labelled nucleoside mix from Saccharomyces cerevisiae RNA as an internal standard (SIL-IS) to a final concentration of 4 ng/µl for the sample RNA and 2 ng/µl for the SIL-IS. For the analysis, 10 µl of the before mentioned mixture were injected into the LC–MS/MS machine. Generation of technical triplicates was obligatory. All mRNA samples were analyzed in biological triplicates. LC separation was performed on an Agilent 1200 series instrument, using 5 mM ammonium acetate buffer as solvent A and acetonitrile as buffer B. Each run started with 100% buffer A, which was decreased

to 92% within 10 min Solvent A was further reduced to 60% within another 10 min Until minute 23 of the run, solvent A was increased to 100% again and kept at 100% for 7 min to re-equilibrate the column (Synergi Fusion, 4 µM particle size, 80 Å pore size, 250 × 2.0 mm, Phenomenex, Aschaffenburg, Germany). The ultraviolet signal at 254 nm was recorded via a DAD detector to monitor the main nucleosides. MS/MS was then conducted on the coupled Agilent 6460 Triple Quadrupole (QQQ) mass spectrometer equipped with an Agilent JetStream ESI source which was set to the following parameters: gas temperature, 350 °C; gas flow, 8 l/min; nebulizer pressure, 50 psi; sheath gas temperature, 350 °C; sheath gas flow, 12 l/min; and capillary voltage, 3,000 V. To analyze the mass transitions of the unlabelled m6A and all 13C m6A simultaneously, we used the dynamic multiple reaction monitoring mode.

**Analysis of m6A methylation by TLC**. Total RNA was extracted with Trizol (Invitrogen) and PolyA mRNA from oligo dT selection was prepared according to the manufacturer (Promega). For each sample, 50 ng of polyA mRNA was cut with RNAse T1 (1000U, Fermentas) and 5′-endlabeled using 10 U of T4 PNK (NEB) and 0.5 µl [γ-$^{32}$P] ATP (6000 Ci/mmol, 25 µM; Perkin-Elmer) in T4 PNK buffer for 2 h. The labelled RNA was precipitated, washed twice with 70% ethanol, resuspended in 10 µl of 50 mM sodium acetate buffer (pH 5.5) and digested with P1 nuclease (SIGMA) for 1 h at 37 °C. Two microliters of each sample was loaded on cellulose F TLC plates (20 × 20 cm; Merck) and run in a solvent system of isobutyric acid:0.5 M NH$_4$OH (5:3, v/v), as first dimension, and isopropanol:HCl:water (70:15:15, v/v/v), as the second dimension. TLCs were repeated from biological replicates. The identity of the nucleotide spots was determined as described[14]. For the quantification of spot intensities on TLCs, a storage phosphor screen (K-Screen; Kodak) and Molecular Imager FX in combination with QuantityOne software (BioRad) were used.

**Immunostaining**. For staining of Drosophila S2R+ and BG3 cells, cells were transferred to the eight-well chambers (Ibidi) at a density of $2 × 10^5$ cells per well. After 30 min, cells were washed with 1× DPBS (Gibco), fixed with 4% formaldehyde for 10 min, and permeabilized with PBST (0.2% Triton X-100 in PBS) for 15 min. Cells were incubated with mouse anti-Myc (1:2000, Enzo, 9E10) or mouse anti-Flag (1:1000) in PBST supplemented with 10% donkey serum overnight at 4 °C. Cells were washed three times for 15 min in PBST and then incubated with secondary antibody and 1× DAPI solution in PBST supplemented with 10% donkey serum for 2 h at 4 °C. After three 15-min washes in PBST, cells were imaged with a Zeiss LSM 710 confocal microscope using a 63× oil immersion objective.

For polytene chromosome staining, the long Hakai isoform was expressed in salivary glands with C155-GAL4 as described[14]. Briefly, larvae were grown at 18 °C under non-crowded conditions. Salivary glands were dissected in PBS containing 4% formaldehyde and 1% TritonX100, and fixed for 5 min, and then for another 2 min in 50% acetic acid containing 4% formaldehyde, before placing them in lactoacetic acid (lactic acid:water:acetic acid, 1:2:3). Chromosomes were then spread under a siliconized cover slip and the cover slip removed after freezing. Chromosome were blocked in PBT containing 0.2% BSA and 5% goat serum and sequentially incubated with primary antibodies (mouse anti-Pol II H5 IgM, 1:1000, Abcam, and rat anti-HA MAb 3F10, 1:50, Roche) followed by incubation with Alexa488- and/or Alexa647-coupled secondary antibodies (Molecular Probes) including DAPI (1 µg/ml, Sigma).

**Co-immunoprecipitation assay and western blot analysis of the Drosophila MACOM subunits**. For the co-immunoprecipitation assays shown in Fig. 1e, f, different combinations of vectors with the indicated tags were co-transfected in S2R+ cells. Forty-eight hours after transfection, cells were collected, washed with DPBS, and pelleted by centrifugation at 400 × g for 10 min The cell pellet was lysed in 1 ml of lysis buffer (50 mM Tris-HCl at pH 7.4, 150 mM NaCl, 0.05% NP-40) supplemented with protease inhibitors and rotated head over tail for 15 min at 4 °C. Nuclei were collected by centrifugation at 1000 × g for 10 min at 4 °C, resuspended in 300 µL of lysis buffer, and sonicated with five cycles of 30 s on and 30 s off at the low power setting. Cytoplasmic and nuclear fractions were joined and centrifuged at 18,000 × g for 10 min at 4 °C to remove the remaining cell debris. Protein concentrations were determined using Bradford reagent (Bio-Rad). For immunoprecipitation, 2 mg of proteins was incubated with 2 µg of anti-Myc antibody coupled to protein A/G magnetic beads (Cell Signalling) in lysis buffer and rotated head over tail overnight at 4 °C. The beads were washed three times for 15 min with lysis buffer, and immunoprecipitated proteins were eluted by incubation in 1× NuPAGE LDS buffer (Thermo Fisher) for 10 min at 70 °C. Eluted immunoprecipitated proteins were removed from the beads, and DTT was added to 10% final volume. Immunoprecipitated proteins and input samples were analyzed by Western blot after incubation for an additional 5 min at 95 °C. For western blot analysis, proteins were separated on a 8% SDS-PAGE gel and transferred to a nitrocellulose membrane (Bio-Rad). After blocking with 5% milk in 0.05% Tween in PBS for 1 h at room temperature, the membrane was incubated with primary antibody in blocking solution overnight at 4 °C. Primary antibodies used were mouse anti-Myc 1:2000 (#9E10, Enzo); mouse anti-HA 1:1000 (#16B12, COVANCE); mouse anti-Tubulin 1:2000 (#903401, Biolegend); mouse anti-Fl(2)d 1:500

(#9G2, DSHB), guinea pig anti-Mettl3 1:500 and rabbit anti-Mettl14 (Lence et al. 2016) 1:250 The membrane was washed three times in PBST for 15 min and incubated for 1 h at room temperature with secondary antibody in blocking solution. Protein bands were detected using SuperSignalWest Pico chemilumi-nescent substrate (Thermo Scientific).

For the co-immunoprecipitation assays, shown in other figures, the following modifications were made:Cells were harvested 72 h after transfection and washed once with DPBS. The cell pellet was lysed in 0.5 ml of NET buffer (50 mM Tris-HCl pH 7.5, 150 mM NaCl, 0.1% Triton and 1 mM EDTA pH 8.0 supplemented with protease inhibitor and 10% glycerol) and sonicated for 3 cycles of 30 s at high power setting followed by incubation with RNase A at ice for 30 min Lysate was centrifuged at $15,000 \times g$ for 10 min at 4 °C. Ten percent of the samples were taken out as input samples and rest were incubated with GFP-magnetic beads on head to toe rotary mixer at 4 °C. The beads were washed three times for 15 min with NET buffer, and proteins were eluted by boiling the beads for 3 min at 95 °C in SDS-page loading dye supplemented with 100 mM DTT. Where applicable, depletion of indicated proteins was performed as described under "Cell culture, RNA interference and transfection". Double amounts of constructs were transfected in several conditions, as indicated in corresponding figure legends.

## Co-immunoprecipitation assay and western blot analysis of the human MACOM subunits

For co-immunoprecipitation assay of the human MACOM subunits, selected truncations of human MACOM components were cloned indi-vidually into pcDNA3-derived vectors with $His_6$-HA3-mCherry- or $His_6$-FLAG3-eGFP- tags. 0.25 mln. HEK293T cells were seeded into 2 ml media in 6-well plates, grown for ~24 h (reaching ~75% confluency) and transfected with corresponding plasmids, 1 µg each, using Xtreme® transfection agent (Roche) at 1:3 ratio. Cells were washed with PBS and harvested ~48 h after transfection using 250 µl of the lysis buffer (20 mM Tris-HCl, pH 7.5, 150 mM KCl, 1 mM EDTA, 0.1% Tween20), supplemented with Roche protease inhibitor. The cells were then flash frozen in liquid nitrogen and stored at −80 °C.

Upon thawing, the cells were further lysed by sonication for 30 s (0.5 s on/2 s off) at 10% amplitude. 2 µg RNase A was added to rule out RNA-mediated interactions. Soluble fraction was separated by centrifugation at full speed for 15 min and incubated with ~18 µl of magnetic anti-FLAG M2 beads (Sigma) for 1 h. The beads were washed 3-4 times with the lysis buffer. The resultant beads were mixed directly with the SDS loading dye and loaded onto 4-20% SDS gels (Biorad). The gels were scanned for GFP and mCherry signals using Typhoon FLA 9500 filters of 473 nm and 532 nm, respectively.

## Immunoprecipitation and ubiquitination analysis of MycGFP-Fl(2)d and MycGFP-Nito

S2R+ cells were transfected with either MycGFP-tagged Nito or Fl(2)d proteins as described above. Forty-eight hours post-transfection attached cells in the 10-cm cell culture dish were washed 2x with cold PBS on ice. Cells were lysed with 1 ml of modified RIPA lysis buffer (50 mM Tris-HCl pH 7.5, 150 mM NaCl, 1 mM EDTA, 1% NP-40, 0,1% Na-deoxycholate), supplemented with complete protease inhibitor cocktail, 5 mM beta-glycerophosphate, 5 mM NaF, 1 mM Na-orthovanadate, 10 mM N-ethylmaleimide. Cells were then collected and incubated for 10 min on ice and centrifuged 15 min at $16,000 \times g$ at 4 °C. Supernatant was transferred to a new tube and protein concentration measured using Bradford. 1.5 mg of proteins was incubated with 20 µL of washed GFPTrap-A beads (Chro-motec) for 1 h at 4 °C end-over-end mixing. Beads were collected by centrifugation (3000 rpm, 1 min) and the supernatant removed. Beads were washed 1× with dilution buffer (10 mM Tris-HCl pH 7.5, 150 mM NaCl, 0.5 mM EDTA, 1× Pro-tease Inhibitor (Sigma), 10 mM N-ethylmaleimide), 3× with stringent wash buffer (8 M Urea, 1% SDS in 1× PBS) and 1× with wash buffer (1% SDS in 1× PBS). Forty-microlitres of 2× LDS sample buffer (Invitrogen) supplemented with 1 mM DTT was added and beads were incubated for 10 min at 70 °C. Twenty-five percent of eluted proteins were analyzed by WB. Primary antibodies used were as follows: mouse anti-Myc 1:2000 (#9E10, Enzo) and mouse Ub antibody (P4D1, Santa Cruz).

The remaining 75% of eluted proteins were used for ubiquitinome and proteome analysis. Proteins were alkylated with 5.5 mM CAA for 30 min at RT in the dark, stained using the Colloidal Blue Staining Kit (Life Technologies) and digested in-gel using trypsin. Peptides were extracted from gel and desalted on reversed-phase C18 StageTips[47]. Samples were then subjected to MS and peptide identification.

## Stable isotope labelling by amino acids in cell culture (SILAC)

For SILAC experiments (ubiquitinome and proteome of Hakai depleted cells), S2R+ cells were grown in Schneider medium (Dundee Cell) supplemented with either heavy (Arg10, Lys8) or light amino acids (Arg0, Lys0) (Sigma) for 6–8 passages and successful incorporation was confirmed by LC–MS/MS.

## Ubiquitylome and proteome analysis in S2R+ cells

Ubiquitinome and pro-teome analysis of control and Hakai depleted SILAC S2R+ cells or Fl(2)d/Vir depleted S2R+ cells was performed as described previously[43,48]. Following mod-ifications were made: S2R+ cells were grown in Schneider medium (Dundee Cell) supplemented with either heavy (Arg10, Lys8) or light amino acids (Arg0, Lys0)

(Sigma) or without supplement for the label-free proteomes (Fl(2)d and Vir KD). Depletion of the different factors was performed with corresponding double-stranded RNA three times during 6 days and scaled up to obtain 50 mg of proteins per replicate (8–10, 15-cm cell culture dishes). Six hours prior to cell lysis, the MG132 proteasome inhibitor was added to a final concentration of 15 µM. Cells were lysed in modified RIPA lysis buffer (50 mM Tris-HCl pH 7.5, 150 mM NaCl, 1 mM EDTA, 1% NP-40, 0.1% Na-deoxycholate) supplemented with complete protease inhibitor cocktail (Roche), 5 mM -glycerophosphate, 5 mM NaF, 1 mM Na-orthovanadate, 10 mM N-ethylmaleimide. 1.5 ml of buffer was used per each 15-cm dish. All lysates of the same transfection were combined in a falcon and protein concentrations were measured by Bradford. Hundred micrograms of each protein sample was collected for WB analysis. For ubiquitylome and proteome analysis 25 mg of heavy and 25 mg of light labelled protein lysates were joined in a 1:1 ratio as follows: for forward experiment heavy labelled lysates with Hakai KD and light labelled lysates with control KD were joined, and vice versa for reverse experiment. 50 µg were then collected for proteome analysis. Finally, ice-cold acetone was added to 80% final conc. (4xV) and precipitated O/N at −20 °C and subjected to MS.

## Label-free proteomes

A total of $15 \times 10^6$ S2R+ cells were transfected with Myc-GFP-tagged HAKAI-long isoform in a 10-cm cell culture dish. In parallel, cells were transfected with Myc-GFP as negative control. After 72 h, the cells were harvested and washed with cold PBS. Cells were lysed with 1 ml of NET buffer (50 mM Tris-HCl pH 7.5, 150 mM NaCl, 0.1% Triton and 1 mM EDTA pH 8.0) and supplemented with 10% Glycerol and Protease inhibitor cocktail (Roche). Lysates were sonicated for 3 cycles of 30 s at a high power setting followed by incubation with RNAse A on ice for 30 min Lysate was centrifuged at $15,000 \times g$ for 10 min at 4 °C. The cleared lysates were incubated with GFP-magnetic beads on a head-to-toe rotary mixer at 4 °C for 2 h. The beads were washed three times for 15 min with NET buffer, and proteins were eluted by incubating at 70 °C in NuPAGE buffer.

## Proteome and ubiquitylome analyses

MS sample preparation, proteome ana-lysis, MS, and peptide identification were performed as described previously in ref. [48]. For peptide identification in SILAC samples, raw data files were analyzed using MaxQuant (development version 1.5.2.8) to calculate the ratios between the different conditions[49]. For the MaxQuant analysis, we used different parameter groups to define group-specific parameters to invert the SILAC ratios in replicate 2 (Reverse). As result of these settings H/L in both experiments represents HAKAI KD/Non-targeting control. The data with all quantified ubiquitylation sites (di-glycine sites) and protein groups are provided as Supplementary_Data_3 (ubi-quitylome analysis) and Supplementary_Data 4 (proteome analysis).

Analysis of label-free samples was performed using default setting for the LFQ analysis in MaxQuant (1.5.2.8) and the Perseus software version 1.5.6.0 to perform calculation of $p$ value and Student's $t$ test[50]. Parent ion and MS2 spectra were searched against a database containing *D. melanogaster* protein sequences obtained from the UniProtKB released in May 2016 using Andromeda search engine[51].

## Yeast-two-hybrid assay (Y2H)

Yeast-two-hybrid assay was performed using *S. cerevisiae* yeast strain [trp1-901, ura3-2,112, ura3-52, his3-200, gal4Δ, gal80Δ, LYS2::GAL1-HIS3, GAL2-ADE2, met2::GAL7-lacZ]. All tested genes (Mettl3, Mettl14, Fl(2)d, Vir, Nito, Flacc, Hakai) were cloned in vectors pGAD424-GW and pGBT9-GW, with Leu2 and Trp1 markers (kindly provided by Helle Ulrich Lab, IMB Mainz) to express all candidates with either the C-terminal Gal4-activation domain or the C-terminal Gal4-DNA binding domain, respectively. Briefly, yeast cells were grown in YPD medium until they reached OD = 0.6. Cells were centrifuged at 3500 rpm for 7 min at RT and washed 1× with water, 1× with 250 ml of SORB (100 mM LiOAc, 10 mM Tris pH 8.0, 1 mM EDTA pH 8.0, 1 M Sorbitol) and 1× with 100 ml of SORB. Pellets were resuspended in 3.6 ml of SORB and 400 ml of ssDNA carrier was added to competent cells. Fifty microlitres ali-quots of cells were mixed with 100 µg of plasmid DNA. 6× volumes of PEG solution (10 mM Tris pH 8.0, 1 mM EDTA pH 8.0, 40 (w/v)-% PEG 3350) were then added to the cell-DNA mixture that was further incubated at RT for 30 min Next, 1/9 of DMSO were added to cells that were subjected to 15 min heat shock at 42 °C. Following 2 min centrifugation at $1968 \times g$ and RT, the cell pellet was resuspended in 500 µL of water and 100 µL of cell solution was plated onto Trp-/Leu- selection agar plates. After 3 days of incubation at 30 °C, 5 colonies of each transformation were resuspended in 500 µL of water and 4 µL were spotted on selection agar plates (Trp-/Leu- and Trp-/Leu-/His-). Plates were imaged in a 24-h interval.

## RNA-seq analysis

Raw data processing, differential expression analysis and splicing analysis was done as described in ref. [24]. All samples from GSE105900 as well as the Hakai sample from GSE158663 were processed at the same time with the same tools and tool versions. The controls used as a reference for the differ-ential splicing and the differential expression analysis are the same. In short the libraries were sequenced on a NextSeq500 with a read length of 85 bp single read and converted to fastq using bcl2fastq (v.2.19) and mapped against Ensembl release 90 of *D. melanogaster* using STAR ([52], v. 2.5.10)). Counts per gene were derived using featureCounts ([53], v. 2.5.1), Differential expression analysis was performed

using Bioconductor/DESeq2 ([54], v1.16.1)) and filtered for an FDR < 5%, default independent filtering was used. Differential splicing analysis was performed using rMATS ([55], v. 3.2.5) and filtered for an FDR < 10%.

**Statistics**. For m⁶A level measurements datasets were compared using two-tailed Student's $t$ test for unequal variances. Fl(2)d levels upon Control, Metll3, Nito, Vir, Flacc and Hakai KD were compared using two-tailed Student's $t$ test for equal variances. Normality was verified and homogeneity of variances was analyzed with Levene's test. Detailed descriptions of confidence intervals, effect sizes, degrees of freedom are shown in the Source data. Statistical tests used for RNAseq data analysis and mass spectrometry analysis are described in detail under relevant sections of "Methods" part.

**Reporting summary**. Further information on research design is available in the Nature Research Reporting Summary linked to this article.

## Data availability

All data needed to evaluate the conclusions in the paper are present in the paper and/or Supplementary Materials. Additional data related to this paper may be requested from the authors. RNA-seq data are available in GEO: GSE158663. Proteomics data are available in PRIDE: PXD022294. Source data are provided with this paper.

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

## Acknowledgements

We thank the Bloomington, Kyoto and Vienna stock Center for fly lines and the *Drosophila* Genomics Resource Center at Indiana University for plasmids and cell lines. We are indebted to S. Hayashi for his effort to recover the *Hakai*[1] allele as well as BestGene and the University of Cambridge Department of Genetics Fly Facility for injections. We thank members of the Roignant and Soller lab for helpful discussion. We thank Christian Renz from the Ulrich group at IMB for sharing reagents and advices for the yeast-two-hybrid assay and Catia Igreja and Heike Budde from late Elisa Izaurralde lab for sharing plasmids used in this study. We thank the IMB Genomics core facility for their helpful support and the use of its NextSeq500 (INST 247/870-1 FUGG). Research in the laboratory of J.-Y.R. is supported by University of Lausanne, the Swiss National Science Foundation (310030_197906), the Deutsche Forschungsgemeinschaft RO 4681/9-1, RO 4681/12-1 and RO 4681/13-1. M.S. is funded by the BBSRC (BB/R002932/1) and the Leverhulme Trust. Research in the M.J. laboratory is supported by the Swiss National Competence Center for Research (NCCR) RNA & Disease. M.K. is supported by EMBO (ALTF 1087-2018) and Human Frontier Science Program (LT000248 2019-L) post-doctoral fellowships. M.J. is an International Research Scholar of the Howard Hughes Medical Institute and Vallee Scholar of the Bert L & N Kuggie Vallee Foundation. P.Beli is supported by the Emmy Noether Program (BE 5342/1-1 and BE 5342/1-2). C.P. in the lab of J.-Y.R. is supported by a Boehringer Ingelheim Fonds fellowship.

## Author contributions

P. Bawankar, T.L., C.P. M.S. and J.-Y.R. conceived the project. P. Bawankar, T.L., C.P. contributed equally and are listed in the alphabetical order. All authors contributed to the experimental design, analysis, and interpretation of results. Experimental contributions were as follows: T.L. initiated the work, which was taken over by P. Bawankar and C.P. They performed all experiments except the following ones: I.U.H., M.P.N. and M.S. performed molecular biology, immunostaining of polytene chromosomes and genetic experiments. M.K. performed the HEK293T co-immunoprecipitation assays with help from M.J. J.B.H. and P. Beli performed MS-based proteomics for ubiquitylome, Hakai interactomes and proteome analysis. D.J., F.M.R. carried out mass spectrometry measurement with the help of M.H. N.K. carried out the bioinformatic analysis. V.M. helped with the generation of *Hakai* mutants. P. Bawankar, T.L., C.P., M.S. and J.-Y.R. wrote the paper with input from all the other authors.

## Competing interests

The authors declare no competing interests
