## [Peer Review File · Nature Communications]

REVIEWER COMMENTS

Reviewer #1 (Remarks to the Author):

The m6A RNA methyltransferase complex containing METTL3 as the catalytic enzyme is a topic of great scientific interest. Previous studies from the authors have shed light on the composition, structure and biology of this complex. The METTL3 complex is composed of several components, including an E3 ubiquitin ligase called Hakai, whose molecular function is poorly understood. Here the authors show that the protein forms part of fly and human complexes, while its depletion in flies result in reduced RNA modification levels and methylation-dependent functions in vivo.

Interestingly, they find that while the ubiquitination activity is not essential for in vivo function, the catalytic domain serves to dimerize the protein. Pointing to its critical role in complex integrity, loss of Hakai results in destabilization of several subunits of this complex, with dramatic consequences for methylation deposition in vivo.

In my opinion, this is the most comprehensive study on the intercomponent interactions of this very important complex, examining it in both flies and human cells. It will be a valuable reference material for the m6A field. I support its immediate publication.

I have only a few minor suggestions for text changes.

1. Figure 4C. Why are there more differentially spliced events in KD of non-catalytic components than KD of the METTL3 itself? Please explain clearly in the text.
2. Page 14. Section title 5 “ ...m6A biogenesis” Instead of biogenesis maybe use “activity”?
3. Figure 1 can definitely use a cartoon with the MACOM complex with names of the fly and human proteins indicated in it. It allows the reader to follow the interactions identified in Fig1-2 and Fig5-6.

Reviewer #2 (Remarks to the Author):

The manuscript, "Hakai is required for stabilization of core components of the m6A mRNA methylation machinery" by Praveen Bawankar et al. is an interesting and important contribution to RNA Biology and of the role of RNA modification in gene regulation. The paper makes several key claims: 1) that Hakai gene product and its mammalian counterpart is a MACOM subunit, 2) the Hakai protein physically interacts with a subset of other MACOM complex members, 3, 4) that Hakai is

important for in vivo m6A methylation as evidenced by reduced and altered methylation of RNA and by altered slicing of the Sxl gene, 5) that Hakai is needed to stabilize other MACOM components and that it does so without need for its reputed E3 ubiquitin ligase activity. The claims are all well supported by the experiments presented and represent a significant gain in our understanding of the role that Hakai plays in RNA methylation.

My only reservations about the paper related to the descriptions and presentation of the genetic experiments which make the arguments hard to follow, obscure some data, and suggest things are simpler than the data allow. The actual experiments are fine. All the following comments relate to the experiments presented in Fig. 3.

The experiment presented in Fig. 3E uses a sensitized genetic background with a low-level of female-specific lethality in which any defects in m6A components would be expected to exacerbate female lethality. This is exactly what is observed with Hakai and with the control Mett13 control. The problem is the simplified description of the logic of the experiment. The female lethality depends on reduced expression of the Sxl isoforms that initiate Sxl autoregulation (caused by both the loss of maternal da, and by the loss of one copy of zygotic Sxl) and by reduced dose of the products that complete and maintain autoregulated splicing (a consequence of Sxlf7bo). I think the solution here is to avoid the details as they distract rather than help. What really matters is that this is genetic system that is expected to make Sxl splicing hypersensitive to reductions in m6 complex function. I also think it would be more clear to drop the mention of dosage compensation. The proximal cause of the lethality is that reduced hakai dose causes a reduction in Sxl expression that leads to female-lethality.

In the experiment in Fig. 3F it is impossible to determine the actual number of flies analyzed. The cross was between hak2/Balancer and Df(hak)/Balancer flies, but the controls are not specified. We are told in the the text that 67 hak2/Df flightless males emerged and that a much smaller, but unspecified (in the text), and illegible (in the figure), number of females emerged. We are thus left with trying to figure how 67 experimental males (with about 90% viability) relates to 339 control flies. To fix the problems, the # of experimental females observed should be stated in the text, and the control flies precisely defined. If my hunch is correct, and the authors did not distinguish the control hak2/Balancer and Df(hak)/Balancer flies, they should define the viability standard as one half the number of heterozygous Balancer-bearing males or females (but not both). That way the reference value would presumably be about 75 flies (67/0.90).

In the description of experiments in Fig. 3G, the vir2F allele is described as a dominant negative. In fact, as experiment in Fig. 3H illustrates, and the published literature shows, the vir2F allele is recessive. What the experiment in Fig. 3G documents is that females bearing only the vir2F allele (vir2f/Df) are inviable but that they are largely rescued if the flies are also heterozygous for hak1. This is a solid result that strengthens the paper, but it's very hard to interpret with respect to

mechanism. What the authors want to argue is that the "dominant negative" [sic] vir2F allele prevents Sxl recruitment (itself a vague notion) and that "weakening" (also vague) the m6A complex allows Sxl better access and thus rescues female viability. It may well be that the mechanism of rescue by reducing hakai dosage is by destabilizing the the vir2F-containing MACOM complex but that is not easily squared with the published finding that reducing the dose of the MAC complex component, mettl3 (lme4), also rescues vir2F-dependent female viability. I think the solution is for the authors is to take the powerful result as a fact, but resist the speculative interpretation.

The experiment in Fig. 3H are powerful and appropriately described. (They also show that vir2F is, at best, sparingly dominant to the partially defective vir2 allele.) I would, however, encourage two modifications to the figure. First, correct the typo on the y-axis "apperence", and second, modify the genotypes so that they more clearly reflect the fact that hakai and vir are on the same chromosome. My suggestion is to label the genotypes of the two right hand sections as + vir / + vir and hak1 vir/+ vir. Doing so provides an additional advantage in that would show how the cross was actually done.

Reviewer #3 (Remarks to the Author):

I don't understand the interpretation of Fig 5A by the authors and, honestly, I think that they may underestimate the value of their data set.

On p14 the authors state: "Cells isotopically labelled with heavy amino acids were depleted for Hakai and cells isotopically labelled with light amino acids served as a control in the forward experiment. A vice versa depletion was performed in the reverse experiment (Supplementary Fig. 8C). [...] We found over 3000 ubiquitination sites, but unexpectedly not a single site was reduced in response to Hakai depletion and only one site in SesB was 1.5-fold increased (Fig. 5A, Supplementary Table 3). Therefore, this experiment strongly suggests that Hakai does not act as a general E3 ubiquitin ligase in Drosophila cells. Alternatively, it is possible that its ubiquitination activity depends on specific external stimuli."

On p32, however, they state that "For ubiquitinome and proteome analysis 50 mg of proteins of heavy and light replicate were joined in a 1:1 ratio as follows: for forward experiment heavy labelled

cells with control KD and light labelled cells with Hakai KD were joined, and vice versa for reverse experiment.”

First of all, the number of diGly peptides obtained from 50 (!) mg of starting material seems somewhat low and at first I was under the impression that the limited coverage may be the problem here. However, upon reanalyzing the source data set using Suppl Table 3, I noticed something else and I think that the x-axis in Fig 5A is mislabeled.

In case of a label swap SILAC experiment (i.e. FWD: H-CTRL, L-Hakai KD vs REV: L-CTRL, H-Hakai KD | this labeling scheme according to the M&M section, p32), the diGly peptides showing consistent up- or downregulation in both duplicates should be searched for in the lower right quadrant, i.e. positive H:L values for the FWD experiment and negative H:L values for the REV experiment. In this case, knock down of a ligase would result in less ubiquitination and therefore H:L ratios >1 in the FWD experiment and H:L ratios <1 in the REV experiment. This is in fact exactly what you see in Fig 5A: the cloud of data points spreading into the lower right quadrant. This cloud is typical for ubiquitinome assays upon e.g. proteasome inhibition or knockout of a ubiquitin ligase. In conclusion, this means that, in contrast to the authors' own conclusion, in fact there seems to be (as expected?) a huge effect on the ubiquitinome upon Hakai KD.

Unless I'm missing something, I would suggest the authors to reanalyze this data set.

We thank the reviewers for their suggestions that help us improving the quality of our manuscript. A point-by-point response addressing each of the comments is presented below.

Overall no major experiments were requested. In Figure 2 we nevertheless add a new information regarding the binding of Human WTAP to VIRMA. We now show that instead of one, two VIRMA domains are involved in the interaction with WTAP, of which one is conserved in *Drosophila*. This new information extends the analysis of our interaction assays and does not modify the overall conclusions of the manuscript.

Reviewer #1 (Remarks to the Author):

The m6A RNA methyltransferase complex containing METTL3 as the catalytic enzyme is a topic of great scientific interest. Previous studies from the authors have shed light on the composition, structure and biology of this complex. The METTL3 complex is composed of several components, including an E3 ubiquitin ligase called Hakai, whose molecular function is poorly understood. Here the authors show that the protein forms part of fly and human complexes, while its depletion in flies result in reduced RNA modification levels and methylation-dependent functions in vivo. Interestingly, they find that while the ubiquitination activity is not essential for in vivo function, the catalytic domain serves to dimerize the protein. Pointing to its critical role in complex integrity, loss of Hakai results in destabilization of several subunits of this complex, with dramatic consequences for methylation deposition in vivo.

In my opinion, this is the most comprehensive study on the intercomponent interactions of this very important complex, examining it in both flies and human cells. It will be a valuable reference material for the m6A field. I support its immediate publication.

I have only a few minor suggestions for text changes.

We like to thank the reviewer for his enthusiasm about our work and to support its publication.

1. Figure 4C. Why are there more differentially spliced events in KD of non-catalytic components than KD of the METTL3 itself? Please explain clearly in the text.

We made similar observations in our previous studies in Lence et al, 2016 and Knuckles et al, 2018. The stronger impact of MACOM components on splicing is consistent with the genetic studies showing that the loss of function of MACOM components induces fly lethality while flies mutant for *Mett3* and *Mettl14* are viable. This indicates that MACOM components bear additional role(s) beyond their involvement within the MAC complex. Whether this function also involves m6A through interaction with another methyltransferase or whether this is an m6A independent function remains to be investigated.

We have now included this paragraph:

Note that depletion of MACOM components has a stronger impact on gene expression and splicing compared to the loss of MAC components. This is consistent with previous genetic data showing that *Mettl3* and *Mettl14* are dispensable for fly viability while MACOM subunits are not, supporting additional function(s) for MACOM components.

2. Page 14. Section title 5 “m6A biogenesis” Instead of biogenesis maybe use “activity”?

We amended the text as followed:

“The Hakai ubiquitination domain but not its activity is required for m6A biogenesis” was replaced by “The Hakai ubiquitination domain but not its activity is required for maintaining MACOM integrity”

3. Figure 1 can definitely use a cartoon with the MACOM complex with names of the fly and human proteins indicated in it. It allows the reader to follow the interactions identified in Fig1-2 and Fig5-6.

As suggested we included a small table in Figure 1 with the names of MAC and MACOM components of both organisms. In addition we added a new Figure 7 with a model for the methyltransferase complex organization and the impact of Hakai on its integrity.

Reviewer #2 (Remarks to the Author):

The manuscript, "Hakai is required for stabilization of core components of the m6A mRNA methylation machinery" by Praveen Bawankar et al. is an interesting and important contribution to RNA Biology and of the role of RNA modification in gene regulation. The paper makes several key claims: 1) that Hakai gene product and its mammalian counterpart is a MACOM subunit, 2) the Hakai protein physically interacts with a subset of other MACOM complex members, 3, 4) that Hakai is important for in vivo m6A methylation as evidenced by reduced and altered methylation of RNA and by altered slicing of the Sxl gene, 5) that Hakai is needed to stabilize other MACOM components and that it does so without need for its reputed E3 ubiquitin ligase activity. The claims are all well supported by the experiments presented and represent a significant gain in our understanding of the role that Hakai plays in RNA methylation.

My only reservations about the paper related to the descriptions and presentation of the genetic experiments which make the arguments hard to follow, obscure some data, and suggest things are simpler than the data allow. The actual experiments are fine. All the following comments relate to the experiments presented in Fig. 3.

We are delighted by the reviewer appreciation of our progress in understanding the m6A mRNA methylation machinery. As detailed below, we have taken care to make the genetic analysis accessible to a broad readership by improving the presentation of the data. In addition, we have put all the details about the genetic crosses and the data from them into a supplementary source data file.

The experiment presented in Fig. 3E uses a sensitized genetic background with a low-level of female-specific lethality in which any defects in m6A components would be expected to exacerbate female lethality. This is exactly what is observed with Hakai and with the control Mett13 control. The problem is the simplified description of the logic of the experiment. The female lethality depends on reduced expression of the Sxl isoforms that initiate Sxl autoregulation (caused by both the loss of maternal da, and by the loss of one copy of zygotc Sxl) and by reduced dose of the products that complete and maintain autoregulated splicing (a consequence of Sxlf7bo). I think the solution here is to avoid the details as they distract rather than help. What really matters is that this is genetic system that is expected to make Sxl splicing hypersensitive to

reductions in m6 complex function. I also think it would be more clear to drop the mention of dosage compensation. The proximal cause of the lethality is that reduced *Hakai* dose causes a reduction in *Sxl* expression that leads to female-lethality.

We have simplified the description of this genetic interaction experiment according to the reviewer suggestion. However, the cause of female lethality is mis-regulation of dosage compensation from reduced *Sxl* levels and we felt that dosage compensation needs to be mentioned in this context to understand the assay. We have tried our best to explain this better.

We have replaced:

To test whether *Hakai* is required for *Sxl* autoregulation, we made use of a genetically sensitized background whereby one copy of *daughterless (da)*, which is involved in *Sxl* transcription, and one copy of *Sxl* required for *Sxl* autoregulation were removed. As for *Mettl3^{null}* mutants, also removal of one copy of *Hakai* killed females (Fig. 3E), due to aberrant dosage compensation.

By

To test whether *Hakai* is required for *Sxl* autoregulation, we made use of a genetically sensitized background based on reduced *Sxl* levels by removal of one copy of *daughterless (da)*, which is involved in *Sxl* transcription, and one copy of *Sxl* required for *Sxl* autoregulation. In the progeny of a cross between *da^{Df/+}; Mettl3^{null/+}* females and *Sxl^{7B0}* null males, most females died (Fig. 3E). Likewise, also removal of one copy of *Hakai* killed females (Fig. 3E).

In the experiment in Fig. 3F it is impossible to determine the actual number of flies analyzed. The cross was between *hak2/Balancer* and *Df(hak)/Balancer* flies, but the controls are not specified. We are told in the text that 67 *hak2/Df* flightless males emerged and that a much smaller, but unspecified (in the text), and illegible (in the figure), number of females emerged. We are thus left with trying to figure how 67 experimental males (with about 90% viability) relates to 339 control flies. To fix the problems, the # of experimental females observed should be stated in the text, and the control flies precisely defined. If my hunch is correct, and the authors did not distinguish the control *hak2/Balancer* and *Df(hak)/Balancer* flies, they should define the viability standard as one half the number of heterozygous *Balancer*-bearing males or females (but not both). That way the reference value would presumably be about 75 flies (67/0.90).

We have now explained the cross in the text and given the numbers. In addition, we have put all the details about the genetic crosses and the data from them into a supplemental file.

We have replaced

Furthermore, when we crossed *Hakai²*, which harbors an early stop codon, to *Df(2L)Exel8041* to normalize genetic background we observed strong female lethality (Fig. 3F). Although the few females we obtained did not show sexual transformation, all male flies were flightless (n=67), as observed for *Mettl3^{null}* and *Mettl14^{null}* mutants

By

Furthermore, when we crossed *Hakai*²/*CyO* females, which harbors an early stop codon, to *Df(2L)Exel8041/CyO* males to normalize genetic background, we observed strong female lethality compared to *CyO* balancer carrying control animals (149 females and 144 males, Fig. 3F). Although the two females we obtained did not show sexual transformation, all male flies were flightless (n=44), as observed for *Mettl3*^{null} and *Mettl14*^{null} mutants.

In the description of experiments in Fig. 3G, the *vir2F* allele is described as a dominant negative. In fact, as experiment in Fig. 3H illustrates, and the published literature shows, the *vir2F* allele is recessive. What the experiment in Fig. 3G documents is that females bearing only the *vir2F* allele (*vir2f/Df*) are inviable but that they are largely rescued if the flies are also heterozygous for *hak1*. This is a solid result that strengthens the paper, but it's very hard to interpret with respect to mechanism. What the authors want to argue is that the "dominant negative" [sic] *vir2F* allele prevents *Sxl* recruitment (itself a vague notion) and that "weakening" (also vague) the m⁶A complex allows *Sxl* better access and thus rescues female viability. It may well be that the mechanism of rescue by reducing *hakai* dosage is by destabilizing the the *vir2F*-containing MACOM complex but that is not easily squared with the published finding that reducing the dose of the MAC complex component, *mettl3* (*lme4*), also rescues *vir2F*-dependent female viability. I think the solution is for the authors is to take the powerful result as a fact, but resist the speculative interpretation.

Indeed, the reviewer is correct that per definition, *vir2F* is recessive and we have removed the notion that the *vir2F* allele is dominant negative as we don't see any signs of sexual transformation in *vir2F/+* females.

We have replaced:

To further confirm the involvement of *Hakai* in *Sxl* alternative splicing we made use of the dominant negative *vir*^{2F} allele, that prevents *Sxl* recruitment and results in female lethality (Hausmann et al. 2016). We found that weakening the m⁶A complex by removal of one copy of *Hakai* restored female viability of *vir*^{2F}/*Df(2R)BSC778* females by correcting *Sxl* alternative splicing (Fig. 3G).

by

To further confirm the involvement of *Hakai* in *Sxl* alternative splicing we made use of the female-lethal *vir*^{2F} allele (Hausmann et al. 2016). We found that removal of one copy of *Hakai* restored female viability of *vir*^{2F}/*Df(2R)BSC778* females by correcting *Sxl* alternative splicing (Fig. 3G), as shown previously for other components of the methyltransferase complex.

The experiment in Fig. 3H are powerful and appropriately described. (They also show that *vir2F* is, at best, sparingly dominant to the partially defective *vir* allele.) I would, however, encourage two modifications to the figure. First, correct the typo on the y-axis "apperence", and second, modify the genotypes so that they more clearly reflect the fact that *hakai* and *vir* are on the same chromosome. My suggestion is to label the genotypes of the two right hand sections as + *vir* / + *vir* and *hak1 vir/+ vir*. Doing so provides an additional advantage in that would show how the cross was actually done.

According to the reviewer suggestions, we have corrected the label of the Y-axis in Fig 3H and modified the genotypes in Fig 3H and 3L-N.

Reviewer #3 (Remarks to the Author):

I don't understand the interpretation of Fig 5A by the authors and, honestly, I think that they may underestimate the value of their data set.

We thank the reviewer for the careful review of our manuscript and for pointing out the discrepancy in the presented data. We incorrectly described the experimental set-up in the material and methods section of the manuscript but it was correctly described in the main text.

On p14 the authors state: "Cells isotopically labelled with heavy amino acids were depleted for Hakai and cells isotopically labelled with light amino acids served as a control in the forward experiment. A vice versa depletion was performed in the reverse experiment (Supplementary Fig. 8C). [...] We found over 3000 ubiquitination sites, but unexpectedly not a single site was reduced in response to Hakai depletion and only one site in SesB was 1.5-fold increased (Fig. 5A, Supplementary Table 3). Therefore, this experiment strongly suggests that Hakai does not act as a general E3 ubiquitin ligase in *Drosophila* cells. Alternatively, it is possible that its ubiquitination activity depends on specific external stimuli."

On p32, however, they state that "For ubiquitinome and proteome analysis 50 mg of proteins of heavy and light replicate were joined in a 1:1 ratio as follows: for forward experiment heavy labelled cells with control KD and light labelled cells with Hakai KD were joined, and vice versa for reverse experiment."

First of all, the number of diGly peptides obtained from 50 (!) mg of starting material seems somewhat low and at first I was under the impression that the limited coverage may be the problem here. However, upon reanalyzing the source data set using Suppl Table 3, I noticed something else and I think that the x-axis in Fig 5A is mislabeled.

We fully agree with the reviewer that we cannot exclude to have not quantified the ubiquitylated sites that are regulated by Hakai due to the limited depth of the analysis. We now tone down our statement and address this point at pages 14-15:

Previous text:

"Therefore, this experiment strongly suggests that Hakai does not act as a general E3 ubiquitin ligase in *Drosophila* cells. Alternatively, it is possible that its ubiquitination activity depends on specific external stimuli".

Rephrased text:

"Therefore, this experiment suggests that Hakai does not act as an E3 ubiquitin ligase in *Drosophila* S2R+ cells. Alternatively, it is possible that its ubiquitination activity depends on specific external stimuli or that we have not quantified the ubiquitination sites that are regulated by Hakai due to the limited depth of the analysis."

In case of a label swap SILAC experiment (i.e. FWD: H-CTRL, L-Hakai KD vs REV: L-CTRL, H-Hakai KD | this labeling scheme according to the M&M section, p32), the diGly peptides showing consistent up- or downregulation in both duplicates should be searched for in the lower right quadrant, i.e. positive H:L values for the FWD experiment and negative H:L values for the REV experiment. In this case, knock down of a ligase would result in less ubiquitination and therefore H:L ratios >1 in the FWD experiment and H:L ratios <1 in the REV experiment. This is in fact exactly what you see in Fig 5A: the cloud of data points spreading into the lower right quadrant. This cloud is typical for ubiquitinome assays upon e.g. proteasome inhibition or knockout of a ubiquitin ligase. In conclusion, this means that, in contrast to the authors' own conclusion, in fact there seems to be (as expected?) a huge effect on the ubiquitinome upon Hakai KD.

Unless I'm missing something, I would suggest the authors to reanalyze this data set.

We thank the reviewer for pointing out the inconsistency in the description of experimental set-up and we wish to apologise for this mistake. For the ubiquitylome and proteome experiments we have performed a double labelling SILAC experiment (using light and heavy isotope labelled cells) that included a label-swap as correctly described in main text (p14) but wrongly described in the materials and methods section. We have now corrected the following text on p32 (materials and methods):

Previous text:

"For ubiquitinome and proteome analysis 50 mg of proteins of heavy and light replicate were joined in a 1:1 ratio as follows: for forward experiment heavy labelled cells with control KD and light labelled cells with Hakai KD were joined, and vice versa for reverse experiment."

Corrected text:

"For ubiquitylome and proteome analysis 25 mg of heavy and 25 mg of light labelled protein lysates were joined in a 1:1 ratio as follows: for forward experiment heavy labelled lysates with Hakai KD and light labelled lysates with control KD were joined, and vice versa for reverse experiment."

For further clarification of double labelling SILAC experiment, we have now included a table in the Supplemental_Fig_S8C, depicting experimental set-up for replicate 1 (Fwd) and replicate 2 (Rev). Supplemental_Fig_S8C now shows validation of Hakai KD (top) and experimental set-up (below).

Supplementary_Fig_S8C:

Relative expression levels of Hakai are shown as a validation of its KD efficiencies for the ubiquitylome and proteome in S2R+ cells. The mean with standard deviation of three technical measurements is shown (top).

(bottom) Experimental set-up of double labelling SILAC experiment for replicate 1(Fwd) and replicate 2 (Rev).

Experiment	Heavy labelled cells	Light labelled cells
Replicate 1 (Fwd)	Hakai KD	Control
Replicate 2 (Rev)	Control	Hakai KD

For clarity, we generated new Supplemental Table3 (ubiquitylome analysis) and 4 (proteome analysis), where we now indicate the ratios not as heavy to light, but as Hakai KD versus Control KD. Accordingly, we provide new graphs in Figure 5A (ubiquitylome) and Figure 6A (proteome) displaying the corresponding data. Please note that in the replicate 2 (Rev) the ratios were inverted (1/ratio) so that in both replicate experiments ubiquitylation sites that are downregulated represent putative Hakai ubiquitylation sites. We have now included this explanation in the method section.

Previous text in Materials and Methods (p34):

“For peptide identification in SILAC samples, raw data files were analyzed using MaxQuant (development version 1.5.2.8) to calculate the ratios between the different conditions (Cox and Mann 2008).”

Rephrased text Materials and Methods (p34):

“For peptide identification in SILAC samples, raw data files were analyzed using MaxQuant (development version 1.5.2.8) to calculate the ratios between the different conditions (Cox and Mann 2008). The ratios in the replicate 2 (Rev) were inverted (1/ratio). The tables with all quantified ubiquitylation sites (di-glycine sites) and protein groups are provided as Supplemental_Table_3 (ubiquitylome analysis) and Supplemental_Table 4 (proteome analysis).”

The lysates that were used for ubiquitylome analysis were the same as the one used for the proteome analysis where we see downregulation of Vir and Flacc in the Hakai KD cells. In the proteome analysis, we see downregulation of Hakai in replicate 1 (SILAC ratio = 0.41) but was unfortunately not quantified in replicate 2. We wish to note that upon re-inspection of our data, we found a typo mistake in the graph and Heatmap of Fig 6A. The Heatmap was manually created to depict the Fold change (Hakai KD/Control KD) of all MAC, MACOM, YTH proteins and proteins below the defined threshold (as shown by red dashed line in the graph). The value for Hakai was correctly included for replicate 1 (0.41), but was wrongly typed as 0.40 for replicate 2. We wish to apologise for this mistake, which has now been corrected in both the Heatmap as well as in the graph.

REVIEWER COMMENTS

Reviewer #2 (Remarks to the Author):

The revised manuscript "Hakai is required for stabilization of core components of the m6A mRNA methylation machinery" represents an important and interesting contribution to the field of RNA Biology. All of my previous comments, as well as those of the other reviewers, have been satisfactorily addressed.

Reviewer #3 (Remarks to the Author):

REVIEWER #3 | Remarks to the rebuttal letter and revised manuscript

The authors acknowledge that there was a discrepancy in the main text description of the SILAC ubiquitylation assay and the text in the Experimental section. However, they have now taken the setup of the experiment as originally described in the main text as the correct one, whereas my impression was that the description of the experiment in the original experimental section was the correct one, while the description in the main text was incorrect. As a result, the supporting data, the interpretation by the authors and the message of Fig. 5A are in fact largely unchanged as compared to the original submission.

In the source data for this figure, which is the MaxQuant output data file that serves as the basis for Supplementary Table 3, the authors have now added the following note: "[...] *for the MaxQuant analysis we inverted the labels in the Software in a way that both replicates are treated as equally labeled and can be directly compared. Therefore in both, the Fwd and Rev experiment, H represents Hakai KD and L represents Control KD. H/L ratio indicates Hakai KD/Control KD [...]*". It is unclear to me how the settings in the software were adapted exactly (were different parameter groups used?) and which effect this has on the resulting output data. In addition, however, the authors mention in the Experimental Section that "[...] *the ratios in the replicate 2 (Rev) were inverted (1/ratio) [...]*". If both statements were true and the inversion was carried out after this modified MaxQuant analysis, the resulting numbers would be incorrect. Either one of these correction methods should be applied, not both.

Based on the above, I still think the data were misinterpreted. The scatter plot in Fig. 5a shows a large inconsistency between the two replicates (i.e., highly positive H:L ratios in replicate #1, highly negative H:L ratios in replicate #2 and vice versa), which still makes me wonder about the correct interpretation of the data. How do the authors otherwise explain that virtually every diGly peptide shows an inconsistent H:L ratio in these two replicate experiments?

Unfortunately, my initial concerns have not been taken away with this new analysis of the data by the authors.

REVIEWER #3 | Remarks to the rebuttal letter and revised manuscript

The authors acknowledge that there was a discrepancy in the main text description of the SILAC ubiquitylation assay and the text in the Experimental section. However, they have now taken the setup of the experiment as originally described in the main text as the correct one, whereas my impression was that the description of the experiment in the original experimental section was the correct one, while the description in the main text was incorrect. As a result, the supporting data, the interpretation by the authors and the message of Fig. 5A are in fact largely unchanged as compared to the original submission.

In the source data for this figure, which is the MaxQuant output data file that serves as the basis for Supplementary Table 3, the authors have now added the following note: “[...] for the MaxQuant analysis we inverted the labels in the Software in a way that both replicates are treated as equally labeled and can be directly compared. Therefore in both, the Fwd and Rev experiment, H represents Hakai KD and L represents Control KD. H/L ratio indicates Hakai KD/Control KD [...]”. It is unclear to me how the settings in the software were adapted exactly (were different parameter groups used?) and which effect this has on the resulting output data.

We thank the reviewer for carefully evaluating our data.

For the analysis of raw files deriving from Forward (Replicate 1) and Reverse (Replicate 2) di-gly and total proteome experiment, we used the MaxQuant software version 1.5.2.8. We used the function “Set parameter group” to define “Group-specific parameters”:

Multiplicity: 2

Replicate 1 (Forward)

“Light labels: Arg0/Lys0; Heavy labels: Arg10/Lys8”

Replicate 2 (Reverse)

“Light labels: Arg10/Lys8; Heavy labels: Arg0/Lys0”.

The result of these settings is that in the MaxQuant output tables H/L in both experiments represents Hakai KD/Non-targeting control. As supplement to this letter, we provide the MaxQuant parameter file that shows the exact settings that were used for the original MaxQuant analysis with indicated parameter groups as described above.

In addition, however, the authors mention in the Experimental Section that “[...] the ratios in the replicate 2 (Rev) were inverted (1/ratio) [...]”. If both statements were true and the inversion was carried out after this modified MaxQuant analysis, the resulting numbers would be incorrect. Either one of these correction methods should be applied, not both.

We only inverted the ratios for the Replicate 2 (Reverse) once and this was done during analysis of raw data with MaxQuant. We apologize if stating this multiple times was misleading. To clarify this we changed the text in Methods section to be exactly the same as in the Supplementary tables and read as follows: “For the MaxQuant analysis we used different parameter groups to define group-specific parameters to invert the SILAC ratios in replicate 2 (Reverse). As result of these settings H/L in both experiments represents Hakai KD/Non-targeting control.”

The correct inversion of the ratios is apparent when looking at the evidence table from the total proteome experiment that has been done in parallel and from the same lysates after mixing of the proteins in a 1:1 ratio. The evidence file contains all Hakai quantified peptides from the proteome experiment and is now provided as supplement to this letter. This shows that we quantify the same

Hakai peptide M(ox)TDLGGVGLGLELHK (m/z 519.2765, 3+) in both replicate experiments, and that in replicate 1 H/L ratio is 0.26 and in replicate 2 H/L ratio is 0.11.

Based on the above, I still think the data were misinterpreted. The scatter plot in Fig. 5a shows a large inconsistency between the two replicates (i.e., highly positive H:L ratios in replicate #1, highly negative H:L ratios in replicate #2 and vice versa), which still makes me wonder about the correct interpretation of the data. How do the authors otherwise explain that virtually every diGly peptide shows an inconsistent H:L ratio in these two replicate experiments?

Unfortunately, my initial concerns have not been taken away with this new analysis of the data by the authors.

We are confident that the conclusions derived from the ubiquitin remnant profiling experiments in Hakai KD cells are not misinterpreted. We have re-analyzed the raw data again and came to the same conclusions as reported in the manuscript. Knockdown of Hakai does not perturb the abundance of quantified ubiquitylation sites and in this case we do not expect to have any di-glycine-modified peptides whose ratio decreases in both replicates. There are different explanations for this and we also added those in the revised manuscript: (i) Hakai is not an active E3 ligase or is not active under basal conditions in which the experiment has been performed (ii) Ubiquitylation sites that are modified by Hakai are low abundant or generate too long peptides after trypsin digestion and therefore were not quantified.

From our experience, in cases where there is no broad perturbation of ubiquitylation site abundance, deviations during sample processing such as slightly unequal mixing of proteins from light and heavy-labeled cells can lead to systematic errors. In addition, in these type of experiments we frequently observe systematic errors in SILAC quantification likely due to overlapping isotope envelopes from co-eluting peptides, which can lead to apparent correlation.

For these reasons, we performed the label switch experiment that permits to overcome these limitations of SILAC and to identify “real” physiological changes in ubiquitylation site abundance even in cases where there is an effect on only a very minor fraction of cellular ubiquitylation sites. This SILAC-based quantification issue is in particular relevant when the quantification is based on single peptides such as in this case for di-glycine modified peptides or for instance phosphopeptides.

Indeed, if we filter for sites that show decreased ubiquitylation in both replicates ($H/L < 0.75$ in Replicate 1 and 2) we identify proteins that we think might be relevant Hakai substrates such as Ub E3 ligase NEDD4, 26S proteasome non-ATPase regulatory subunit 9 CG9588 (PSMD9 in human) and NEDD8 E1 activating enzyme UBA3. UBA3 is the only NEDD8 E1 activating enzyme, and HAKAI was in Liu et al (PMID: 30041665) suggested to modify substrates with Ub-like modifier NEDD8. Therefore, we think that we identified a few sites that might be physiologically relevant and suggest a role of Hakai in NEDDylation pathway, but these were only slightly decreased in abundance and did not pass the threshold of being 2-fold upregulated in both replicates. For this reason, we do not want to include this data in the current manuscript but we plan to further investigate this possibility in our future work.

REVIEWERS' COMMENTS

Reviewer #4 (Remarks to the Author):

I have focused my review on the proteomics data for the manuscript from Bawankar et. al. It is unfortunate that there is quite some confusion about the process by which the authors have specified the MaxQuant parameters for the SILAC experiment, particularly in that the output files do not clearly reflect this.

Nevertheless, the results of the whole proteome HAKAI-KD and the ubiquitin-GlyGly enrichment look to be processed appropriately. The Gly-Gly experimental data appears to be noisy, but importantly, the observations made based on the data match the authors' claims that HAKAI-kd does not clearly affect ubiquitination in the sample. I think the manuscript is cautious about not overstating the experimental results.

In conclusion, I think the proteomic data looks to be fine in the manuscript.